# Role of the SAF-A/HNRNPU SAP domain in X chromosome inactivation, nuclear dynamics, transcription, splicing, and cell proliferation

Judith A. Sharp, Emily Sparago, Rachael Thomas, Kaitlyn Alimenti, Wei Wang, Michael D. Blower*

Department of Biochemistry and Cell Biology, Boston University Chobanian and Avedisian School of Medicine, Boston, Massachusetts, United States of America

* mblower@bu.edu

## Abstract

SAF-A/HNRNPU is conserved throughout vertebrates and has emerged as an important factor regulating a multitude of nuclear functions, including lncRNA localization, gene expression, and splicing. Here we show the SAF-A protein is highly dynamic and interacts with nascent transcripts as part of this dynamic movement. This finding revises current models of SAF-A: rather than being part of a static nuclear scaffold/matrix structure that acts as a stable tether between RNA and chromatin, SAF-A executes nuclear functions as a dynamic protein, suggesting contacts between SAF-A, RNA, and chromatin are more high turnover interactions than previously appreciated. SAF-A has several functional domains, including an N-terminal SAP domain that binds directly to DNA and RNA. Phosphorylation of SAP domain serines S14 and S26 is important for SAF-A localization and function during mitosis, however, whether these serines are involved in interphase functions of SAF-A is not known. In this study we tested for the role of the SAP domain, and SAP domain serines S14 and S26 in X chromosome inactivation, protein dynamics, gene expression, splicing, and cell proliferation. Here we show that the SAP domain, and SAP domain serines S14 and S26 are required to maintain XIST RNA localization and XIST-dependent histone modifications on the inactive X chromosome, to execute normal protein dynamics, and to maintain normal cell proliferation. In addition, we present evidence that a Xi localization signal resides in the SAP domain, enabling SAF-A to engage with the Xi compartment in a manner distinct from other nuclear territories. We found that the SAP domain is not required to maintain gene expression and plays only a minor role in mRNA splicing. We propose a model whereby dynamic phosphorylation of SAF-A serines S14 and S26 mediates rapid turnover of SAF-A interactions with nuclear structures during interphase. Our data suggest that different nuclear compartments may have distinct requirements for the SAF-A SAP domain to execute nuclear functions, a level of control that was not previously known

**Data availability statement:** All raw reads and summary files are available in GEO under the accession numbers: GSE277216, GSE277217, GSE277219, GSE277221.

**Funding:** This work was supported by the National Institutes of General Medical Sciences GM144352 to MB and National Institutes of General Medical Sciences GM122893 to MB. https://www.nigms.nih.gov/he funders had no role in study design, data collection and analysis, decision to publish, or preparation of the manuscript.

**Competing interests:** The authors have declared that no competing interests exist.

## Author summary

Proper regulation of gene expression is critical for normal organismal development and control of cell growth. Defects in gene expression lead to a spectrum of developmental defects in humans. Gene expression is controlled through both primary DNA sequence as well as various factors that interact with DNA. An important and understudied factor regulating gene expression is the interaction of various types of RNA with chromatin. SAF-A/HNRNPU is a highly abundant nuclear protein that contains both RNA and DNA binding domains and has been hypothesized to tether RNA to chromatin. SAF-A is an essential gene and spontaneous heterozygous mutations in humans lead to neurodevelopmental defects and epilepsy. In this work we use genetically engineered human cell lines to explore the function of the SAF-A SAP domain. We find that the SAF-A SAP domain is important for controlling some aspects of SAF-A function and for access to specific nuclear compartments. Our broad phenotypic analysis provides insight into the essential functions of SAF-A and may illuminate the causes of a rare human disease.

## Introduction

Recent work has shown that RNA plays an active role in forming compartments in the nucleus to regulate various essential functions, such as RNA processing and gene expression [1–5]. RNA and RNA-binding proteins are concentrated near active chromatin and act to organize a network of transcription factors and other chromatin regulators. The proper assembly of these structures is therefore important for essential nuclear functions and to prevent neurological disease [6,7].

The SAF-A/HNRNPU protein is a highly abundant nuclear RNA-binding protein that is essential in mice [8,9] and cancer cells [10]. Mutations in SAF-A in humans or mice lead to a variety of neurological defects [11–13]. SAF-A was originally described as a factor that binds to hnRNA and attaches RNA to nuclear scaffold attachment regions [14–16]. Since then, SAF-A has been implicated in regulation of mRNA splicing [8,11,12,17], gene expression [11,12], nuclear structure [18, 19], and tethering of the XIST and FIRRE lncRNAs to chromatin [20–23]. The SAF-A protein has several conserved functional domains, including an N-terminal SAP domain, a D/E-rich acidic region, a SPRY domain of unknown function, a central ATPase domain, and a C-terminal RGG domain involved in binding RNA and DNA (Fig 1A; [24]). At present, very little is known about which functional domains of SAF-A are required for each cellular process.

The SAP domain is a nucleic acid binding domain present in many factors that regulate aspects of chromosome metabolism [25]. The SAF-A SAP domain binds directly to SAR DNA *in vitro* [26,27]. In addition, recent work has indicated that the SAF-A SAP domain can also bind directly to RNA both *in vitro* [28] and *in vivo* [29]. Two conserved serines present in the SAP domain, S14 and S26, are positioned in the DNA binding

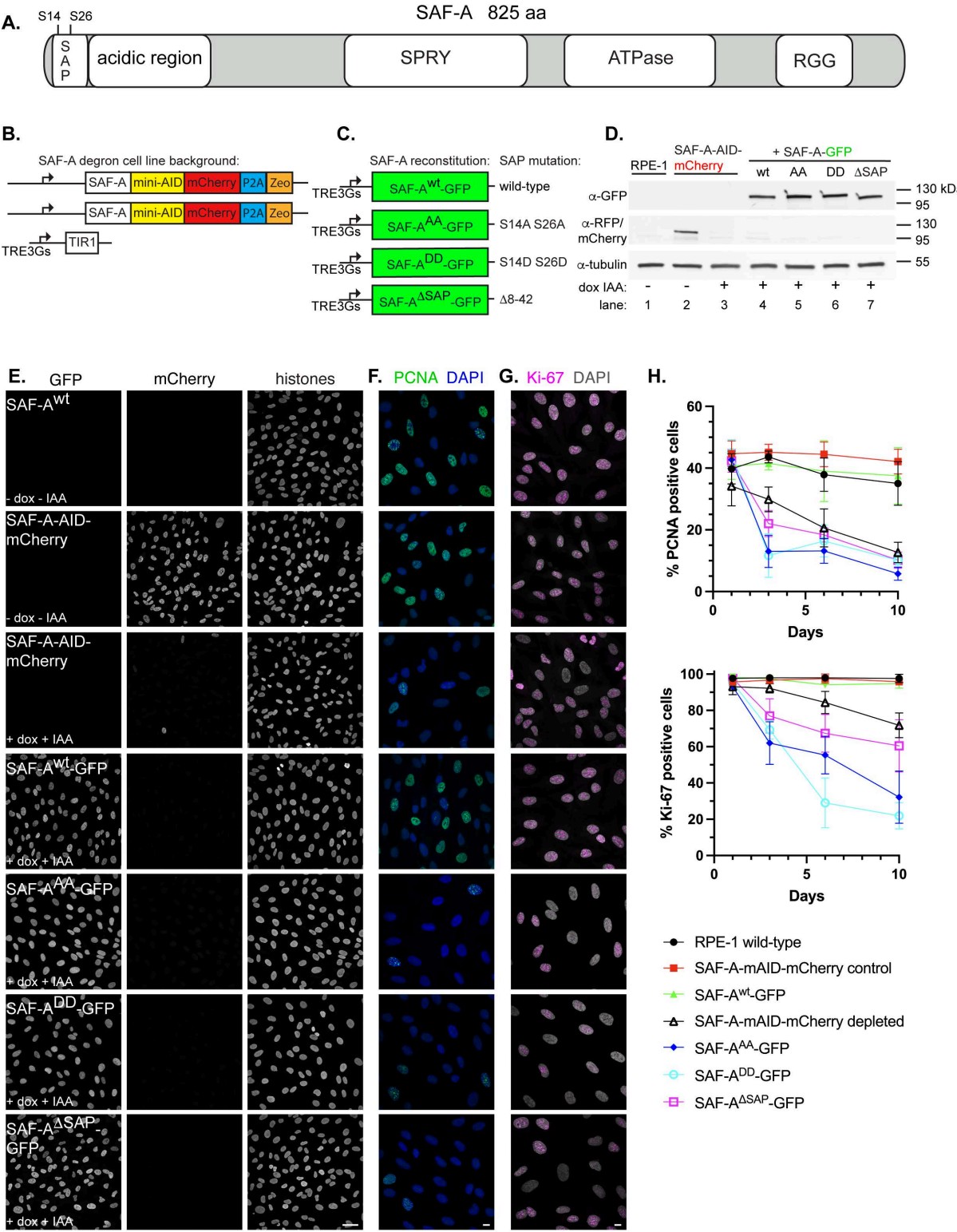

**Fig 1. The SAP domain and phosphorylatable SAP domain serines S14 and 26 are required for cell proliferation.** A. Schematic depicting full-length SAF-A (isoform a, 825 amino acids) with domains drawn to scale. B. Design of the SAF-A-AID-mCherry degron to replace the endogenous SAF-A genes in RPE-1 cells. C. SAF-A transgenes with alanine (phosphomutant) or aspartic acid (phosphomimetic) mutations at positions S14 and S26, or

expressing a SAP domain deletion. D. Western blot analysis of SAF-A cell lines to monitor expression of SAF-A-AID-mCherry (α-RFP panel) and SAF-A-GFP (α-GFP panel) in cell lines. Tubulin was used as a loading control. Lane 1: untagged RPE-1 parental cell line. Lane 2: SAF-A degron cell line, either without (lane 2), or with (lane 3) doxycycline and IAA. Lanes 4-7: cells expressing SAF-A-GFP transgenes, treated with doxycycline and IAA. Molecular weight markers are indicated to the right of the panel. E. Immunofluorescence of cells expressing tagged SAF-A, as indicated by the panel insets. Treatment with doxycycline and IAA as indicated led to complete depletion of SAF-A-AID-mCherry in the vast majority of cells. Integrated SAF-A-GFP transgenes showed uniform expression in the cell population after induction. Bar, 40 mm. F-G. PCNA and Ki-67 immunofluorescence to monitor cell proliferation. Cells were treated with doxycycline and IAA to induce endogenous SAF-A depletion and SAF-A-GFP transgene expression, and were compared to control cell populations. Cells were treated for 1, 3, 6, or 10 days as indicated. SAF-A expression is the same as the panel insets shown in E. All images in panels E-G are rendered as a maximum projection of a 3D stack. Bar, 10 mm. H. Quantitation of cell populations with PCNA- and Ki-67-positive cells expressed as percent of the total population. 300 cells were scored for PCNA and Ki-67 immunofluorescence in two biological replicates; the average and standard deviation is shown.

interface such that the electrostatic potential of the SAP domain is determined by the phosphorylation state of these two serines [27]. As a result, the increased negative charge associated with SAP domain S14 and S26 phosphorylation greatly reduces the affinity of SAF-A binding to DNA relative to the unphosphorylated protein. We previously demonstrated that SAP domain S14 and S26 phosphorylation by Aurora-B plays a critical role in regulating the interaction of SAF-A RNP complexes with the surface of mitotic chromosomes [27]. Since phosphopeptide evidence demonstrates that S14 and S26 are modified in interphase cells [30], we aimed to determine whether S14 and S26 phosphorylation regulates SAF-A nuclear functions.

In this study we present phenotypic analysis of cell lines either lacking SAF-A expression, or expressing a SAP domain allelic series, including mutations which mimic the phosphorylated and unphosphorylated state, and a SAP domain deletion. We found that the SAP domain and phosphorylatable serines S14 and S26 were important for SAF-A function in XIST RNA localization and activity, SAF-A dynamic nuclear mobility, interaction with nascent RNA, and cellular proliferation. In contrast, the contribution of SAF-A to gene expression and mRNA splicing is largely independent of the SAP domain. Taken together, our data suggest dynamic phosphorylation of SAP domain serines S14 and S26 impacts a subset of SAF-A nuclear functions during interphase.

## Results

### The SAF-A SAP domain is required for cell proliferation

To investigate SAF-A nuclear functions in human cells, we first designed a cell line to degrade endogenous SAF-A in diploid, karyotypically stable human RPE-1 cells immortalized with hTERT ( [27]; Fig 1B). Both SAF-A genes were modified at the 3' end to include the minimal auxin-inducible degron sequence and a mCherry fluorescent tag (SAF-A-AID-mCherry). A doxycycline-inducible TIR1 gene encoding the E3 ligase required for auxin-mediated protein degradation was integrated by lentivirus. Control experiments demonstrated efficient depletion of the SAF-A degron allele, resulting in complete depletion of SAF-A-AID-mCherry in ≥ 98% of cells after 24 hours of adding doxycycline and 3-indole acetic acid (IAA; Figs 1D, 1E, and S1A–S1C). Quantitative western blot analysis confirmed depletion of the degron allele (S1B and S1C Fig). We observed that untreated SAF-A-AID-mCherry cells had reduced expression relative to the level of endogenous SAF-A in RPE-1 cells; however, this decrease in expression did not alter cell growth and resulted in very few changes in gene expression compared to RPE-1 cells (S1D–S1F Fig).

To evaluate the role of the SAP domain in SAF-A function, we then integrated doxycycline-inducible GFP-tagged rescue constructs encoding either full-length, wild-type SAF-A (SAF-A$^{wt}$-GFP), the phosphomutant S14A S26A (SAF-A$^{AA}$-GFP), the phosphomimetic allele S14D S26D (SAF-A$^{DD}$-GFP), or a deletion of the SAP domain (SAF-A$^{ΔSAP}$-GFP). In this strategy, the addition of doxycycline and IAA triggers the simultaneous degradation of endogenous SAF-A-AID-mCherry and induction of GFP-tagged SAF-A rescue constructs. Analysis of SAF-A protein levels in these cell lines confirmed SAF-A-GFP expression and degradation of endogenous SAF-A-mAID-mCherry in cell extracts (Fig 1D lanes 4–7) and cell cultures (Fig 1E) under these conditions. Quantitative analysis of expression levels of rescue constructs revealed that they are all expressed within a 2-fold difference of endogenous levels (S1B, S1C, and S1G Fig).

SAF-A is an essential gene required for cell proliferation and embryonic viability [8,9]. We therefore tested whether our tagged SAF-A alleles supported essential functions of SAF-A by monitoring the number of cycling cells over 10 days, using PCNA and Ki-67 as markers for cell proliferation (Fig 1F–1H). Both SAF-A-mAID-mCherry and SAF-A^wt-GFP maintained levels of PCNA- and Ki-67-positive cells comparable to wild-type RPE-1 cells, demonstrating that the tagged alleles encode functional proteins that complement the unmodified SAF-A gene and support essential functions of SAF-A. In contrast, SAF-A depletion caused a significant reduction in cycling cells over the time course of the experiment, as did the modification of SAP domain serines (SAF-A^AA-GFP and SAF-A^DD-GFP) and a SAP domain deletion (SAF-A^ΔSAP-GFP). We note that during the time course of the experiment, each SAP domain mutation resulted in fewer cycling cells compared to the SAF-A depletion, suggesting that the SAP domain alleles are loss of function mutations that display variable degrees of penetrance. Together, these data suggest that the SAP domain is required for essential functions of SAF-A, and that the two phosphorylatable serines S14 and S26 in the SAP domain are critical for the role of SAF-A in supporting normal levels of cell proliferation.

## The SAF-A SAP domain serines are required for XIST RNA localization to the Xi

Several studies have reported that SAF-A is required for the proper localization of the noncoding XIST RNA to the inactive X chromosome (Xi) in female cells [21–23,31,32]. We note these studies used a variety of methods (siRNA, Cre-Lox recombination, and constitutive CRISPR SAF-A KO) that require several days to completely deplete SAF-A from cells, likely due to the high abundance of SAF-A protein. Since our SAF-A degron cell line resulted in robust depletion of SAF-A within 24 hours of drug treatment (Fig 1), we determined whether the XIST RNA localization defect was present after this more acute period of depletion.

RPE-1 cells are female and harbor an Xi with XIST RNA expression and histone modifications associated with facultative heterochromatin [33]. XIST RNA is present at ~50 copies per nucleus in cell culture models of X inactivation [34–37], yet appears as a "cloud" in the confocal light microscope due to the resolution limit described by the point spread function. To evaluate the role of SAF-A in XIST RNA localization, we developed a quantitative imaging assay designed to count resolvable XIST RNA foci in 3D confocal optical stacks, reasoning that a localization defect would result in an increased number of XIST RNA molecules distal to the Xi, therefore also increasing the number of resolvable XIST RNA foci per nucleus. In this assay, we note we are using standard confocal microscopy and are measuring the total number of XIST RNA foci, not single XIST molecules. Cells were treated for 24 hours to deplete endogenous SAF-A-AID-mCherry and induce the SAF-A-GFP transgenes prior to XIST RNA FISH hybridization. 3D optical stacks were then acquired for ≥ 100 interphase nuclei per genotype in two biological replicate experiments. Images were analyzed in batch to count XIST RNA foci using a script we developed in Fiji software (Materials and methods, S1 and S2 Files). Representative XIST RNA FISH patterns are shown in Fig 2A; quantitative analysis of the two biological replicates is depicted in Fig 2B as a violin superplot [38].

Comparison of native RPE-1 cells (SAF-A^wt) with cells expressing SAF-A^wt-GFP demonstrated an equivalent number of XIST RNA foci per nucleus, demonstrating that the tagged construct complements for the role of SAF-A in XIST RNA localization to the Xi. Both SAF-A^wt and SAF-A^wt-GFP nuclei had relatively few resolvable foci, reflecting the close positioning of XIST RNA molecules within the Xi territory. In contrast, SAF-A depletion resulted in an average 2-fold increase in XIST RNA foci, with many events in the population having ~10–50 XIST RNA foci per cell (Fig 2B). Analysis of XIST RNA levels demonstrated that increased XIST foci did not result from increased expression in SAF-A mutant cell lines (S2A and S2B Fig) These data confirm that SAF-A is required to maintain proper XIST RNA localization during this 24-hour experimental window, a period of time equivalent to one cell cycle in RPE-1 cells.

Analysis of cell lines with SAP domain mutations demonstrated that deletion of the SAP domain caused an increase in XIST RNA foci comparable to complete depletion of the SAF-A protein (Fig 2A and 2B). While the phosphomutant allele SAF-A^AA-GFP caused no detectable XIST RNA phenotype, the phosphomimetic allele SAF-A^DD-GFP resulted in potent

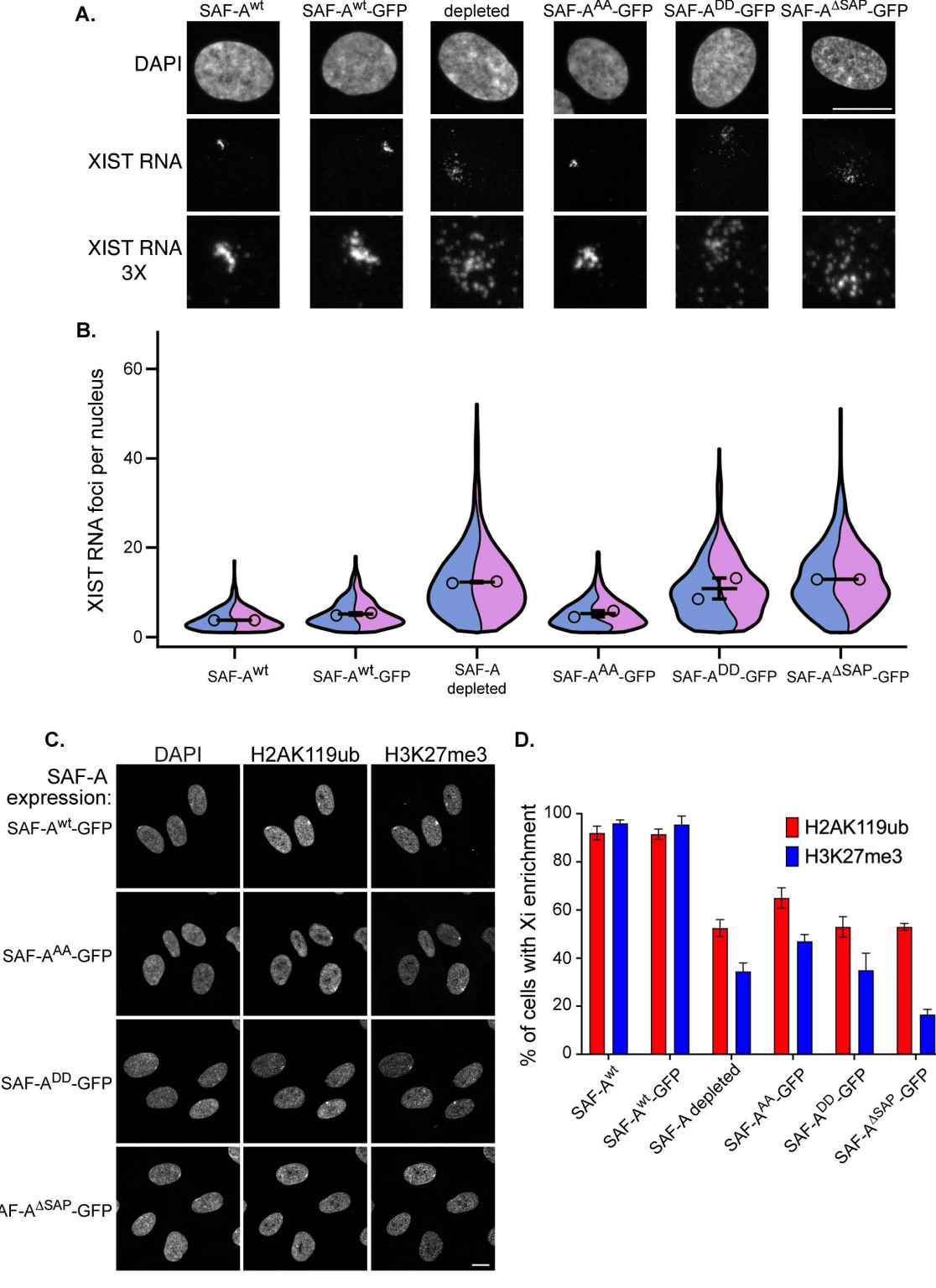

**Fig 2. The SAP domain and SAP domain serines S14 and 26 impact XIST RNA localization and PRC1/PRC2-dependent histone modifications on the Xi.** A. XIST RNA FISH and DAPI staining in RPE-1 cells, and cells expressing SAF-A-GFP transgenes, 24 hours after treatment with doxycy-cline and IAA. Images are rendered as a maximum projection of a 3D stack. Bar, 10 µm. B. Quantitative measurement of XIST RNA foci per cell. The term foci refers to the number of resolvable objects rather than individual molecules. Image stacks were acquired for at least 100 nuclei per genotype

for two biological replicates and analyzed in Fiji software to count XIST RNA foci. Measurements are depicted as violin superplots. The average of each replicate is depicted as an open circle, whereas the average of both replicates is depicted as a horizontal line. The standard deviation of the two averages is shown. Cell n for quantitation of XIST RNA particles is: RPE-1 $n = 193$ and $n = 211$, SAF-A$^{wt}$-GFP $n = 137$ and $n = 236$, SAF-A depleted 1 $n = 122$ and $n = 220$, SAF-A$^{AA}$-GFP $n = 170$ and $n = 206$, SAF-A$^{DD}$-GFP $n = 133$ and $n = 198$, SAF-A$^{\Delta SAP}$-GFP $n = 160$ and $n = 158$. Statistical comparison of number of XIST RNA particles was performed using a one-way ANOVA followed by Tukey's tests with a Bonferroni correction. The $p$-value comparisons between SAF-A$^{wt}$ and all other genotypes are as follows: SAF-A$^{wt}$ versus SAF-A$^{wt}$-GFP, ns. SAF-A$^{wt}$ versus SAF-A depleted, $p = 0.001$. SAF-A$^{wt}$ versus SAF-A$^{AA}$-GFP, ns. SAF-A$^{wt}$ versus SAF-A$^{DD}$-GFP, $p = 0.004$. SAF-A$^{wt}$ versus SAF-A$^{\Delta SAP}$-GFP, $p = 0.001$. C. Immunofluorescence of histone H3K27me3 and H2AK119ub and DAPI staining, 24 hours after treatment with doxycycline and IAA. SAF-A-GFP allele expression is indicated to the left of the panel. Images are rendered as a maximum projection of a 3D stack. Bar, 10 μm. D. Quantitation of cell populations with H3K27me3 and H2AK119ub enrichment on the Xi. 100 cells were scored for H3K27me3 and H2AK119ub enrichment on the Xi in two biological replicates; the average and standard deviation is shown. Statistical comparison was performed using a t-test to determine $p$-values for H2AK119ub enrichment. SAF-A$^{wt}$ versus SAF-A$^{wt}$-GFP, $p = 0.86$, ns. All other comparisons were statistically significant: SAF-A$^{wt}$ versus SAF-A depleted, $p = 0.0065$. SAF-A$^{wt}$ versus SAF-A$^{AA}$-GFP, $p = 0.0174$. SAF-A$^{wt}$ versus SAF-A$^{DD}$-GFP, $p = 0.0084$. SAF-A$^{wt}$ versus SAF-A$^{\Delta SAP}$-GFP, $p = 0.0033$. The same analysis was performed to determine $p$-values for H3K27me3 enrichment. SAF-A$^{wt}$ versus SAF-A$^{wt}$-GFP, $p = 0.8698$, ns. All other comparisons were statistically significant: SAF-A$^{wt}$ versus SAF-A depleted, $p = 0.0019$. SAF-A$^{wt}$ versus SAF-A$^{AA}$-GFP, $p = 0.0021$. SAF-A$^{wt}$ versus SAF-A$^{DD}$-GFP, $p = 0.0069$. SAF-A$^{wt}$ versus SAF-A$^{\Delta SAP}$-GFP, $p = 0.0005$.

XIST RNA mislocalization. These results indicate a critical role for the SAF-A SAP domain serines in maintaining proper XIST RNA localization to the Xi. While the allele modeling unphosphorylated SAP domain serines supports XIST RNA localization, the allele modeling constitutive phosphorylation of the two serines in the SAP domain antagonizes XIST RNA interactions with Xi chromatin.

## The SAF-A SAP domain is required for maintenance of Polycomb-dependent histone modifications on the Xi

XIST RNA triggers the modification of Xi chromatin by recruiting the histone modification enzymes PRC1 to catalyze histone H2AK119ub, and PRC2 to catalyze histone H3K27me3, through a mechanism genetically linked to multiple XIST RNA effector proteins [39]. To investigate whether the SAF-A SAP domain is required to maintain XIST RNA-dependent chromatin modifications on the Xi, we performed immunofluorescence for H2AK119ub and H3K27me3 and scored Xi enrichment in cell populations expressing wild-type and mutant SAF-A-GFP alleles, 24 hours after drug treatment. In both mouse and human cells, H3K27me3 and H2AK119ub form a prominent nuclear body that colocalizes with XIST and the Xi. Comparison of native RPE-1 cell populations (SAF-A$^{wt}$) to those expressing SAF-A$^{wt}$-GFP demonstrated that both genotypes had similar frequencies of H2AK119ub and H3K27me3 Xi enrichment, again demonstrating the functionality of the GFP-tagged protein (Figs 2C, 2D, and S2C). In contrast, SAF-A depletion resulted in markedly decreased H2AK119ub and H3K27me3 Xi enrichment, with approximately half of the cell population having no detectable Xi enrichment of these histone modifications. This is consistent with previous results showing a loss of H3K27me3 on the Xi after siRNA-mediated depletion of SAF-A in mouse cells [21]. We also observed significantly reduced H2AK119ub and H3K27me3 Xi enrichment in cell lines expressing SAF-A$^{AA}$-GFP, SAF-A$^{DD}$-GFP, and SAF-A$^{\Delta SAP}$-GFP. These data demonstrate that the SAF-A SAP domain, and in particular SAP domain serines 14 and 26, are important for maintaining PRC1 and PRC2 functions on the Xi. Further, the data suggest dynamic phosphorylation of the SAF-A SAP domain is important for the role of SAF-A in maintaining XIST RNA-dependent chromatin modifications.

## The SAF-A SAP domain contains a Xi localization signal

Immunofluorescent detection of SAF-A has led to conflicting reports about whether it is enriched or excluded from the Xi chromosome territory [40–42] suggesting that differences in sera or fixation conditions can result in contradictory observations of SAF-A localization. To avoid these potential problems, we performed live imaging of our SAF-A-GFP cell lines to determine whether SAP domain mutations resulted in altered Xi localization. We observed that SAF-A$^{wt}$-GFP was modestly enriched in the Xi chromosome territory as indicated by overlap with the Barr body, consistent with a prior report ([40], Fig 3A). SAF-A$^{AA}$-GFP and SAF-A$^{DD}$-GFP were also present in the Xi chromosome territory, with SAF-A$^{AA}$-GFP

showing modest Xi enrichment. SAF-A$^{DD}$-GFP was present in the Xi territory but not observably enriched. In contrast, SAF-A$^{\Delta SAP}$-GFP was distinctly excluded from the Xi territory. These data show that there is a Xi localization sequence present in the SAF-A SAP domain that is largely independent of serines 14 and 26. It is likely that underrepresentation of SAF-A$^{\Delta SAP}$-GFP on the Xi is an important feature explaining the aberrant XIST RNA localization and Xi chromatin pheno-types associated with this mutation.

**SAF-A exhibits rapid nuclear dynamics and interacts with nascent transcripts**

To test whether the SAP domain plays a role in potentiating SAF-A nuclear dynamics in living cells, we performed FRAP analy-sis in cells expressing SAF-A-GFP transgenes. SAF-A$^{wt}$-GFP exhibited rapid nuclear dynamics such that the $t_{1/2}$ of recovery was an average 2.58 seconds, representing a high degree of nuclear mobility. These rapid nuclear movements suggest that most of the cellular pool of SAF-A participates in high turnover, transient engagements with chromatin or other nuclear structures. Indeed, we note that only an average of 18% of SAF-A$^{wt}$-GFP was estimated to be immobile in multiple experiments imaging cells for a duration of 30 seconds (Fig 3B–3D). Next, we compared SAF-A mobility in cells expressing the SAP domain allelic series: SAF-A$^{AA}$-GFP, SAF-A$^{DD}$-GFP, and SAF-A$^{\Delta SAP}$-GFP. We found that all SAP domain mutations significantly decreased the $t_{1/2}$ of recovery to approximately 2.0 seconds, representing much more rapid nuclear motion than the wild-type protein. These data suggest that the SAP domain is engaging transiently with nuclear structures to impact the nuclear dynamics of SAF-A.

Recent work demonstrated that SAF-A localization and activity is responsive to nuclear transcription [1,19], suggest-ing that SAF-A may interact with nascent transcripts. To determine whether transcription inhibition affects SAF-A mobility, we performed FRAP after inhibition of transcriptional elongation using the CDK9 inhibitor LDC000067 (LDC) [43]. Acute inhibition of CDK9 resulted in a strong decrease in nascent transcription (S3 Fig). Interestingly, SAF-A$^{wt}$-GFP and all SAF-A SAP domain mutants exhibited increased nuclear mobility in the presence of LDC, suggesting that interaction of SAF-A with nascent transcripts is an important determinant of SAF-A nuclear dynamics. SAF-A$^{\Delta SAP}$-GFP had the strongest increase in mobility after transcription inhibition, with SAF-A$^{AA}$-GFP and SAF-A$^{DD}$-GFP having a more modest increase in mobility relative to the SAF-A$^{wt}$-GFP control. The SAF-A immobile fraction was reduced for all genotypes in the presence of LDC, suggesting there is a population of SAF-A molecules that are stably associated with nascent transcripts inde-pendently of the SAP domain (Fig 3D). Taken together, these data demonstrate that SAF-A is a highly dynamic protein that interacts with nascent transcripts and additional nuclear structures through the SAP domain.

Our FRAP data stands in contrast to a prior study which reported much slower rate of SAF-A nuclear motion [40]. We note there are experimental differences between that study and the data presented here. First, those experiments were conducted in transformed, aneuploid HEK293 cells which may have different nuclear properties than the system we describe in RPE-1 cells. In addition, the HEK293 cells were expressing SAF-A-GFP in addition to the endogenous protein, which could affect protein dynamics. We performed controls to ensure that SAF-A$^{wt}$-GFP is the sole source of SAF-A expression in the RPE-1 SAF-A degron cell line, and that the expression level was comparable to untagged native protein levels (S1B, S1C, and S1G Fig). In addition, we have tested our SAF-A$^{wt}$-GFP reagent in multiple genetic assays and found that it complements for normal cell proliferation, normal chromosome segregation, Xi chromatin structure, global RNA transcription, RNA splicing, and cell cycle dependent localization (this study, [27]). Finally, our measured SAF-A dynamics are concordant with those reported for SAF-A expressed from a transgene in RPE-1 cells in a recent preprint [44]. For these reasons, we conclude that the rapid nuclear movements we observe with the SAF-A$^{wt}$-GFP allele are an accurate measurement of normal SAF-A dynamics.

We tested whether SAP domain mutations caused differential interaction with chromatin by analyzing SAF-A immuno-precipitations for the presence of histone H3 (Fig 3E and 3F.) SAF-A$^{wt}$-GFP coprecipitated with histone H3 in a manner that was enriched relative to the no tag and no antibody controls. However, we did not observe significant differences in chromatin interactions for the SAP domain mutants in statistical comparisons to SAF-A$^{wt}$-GFP (Fig 3F). While our live imaging data suggest SAF-A is highly mobile in the nucleus and is contacting nuclear structures transiently, the

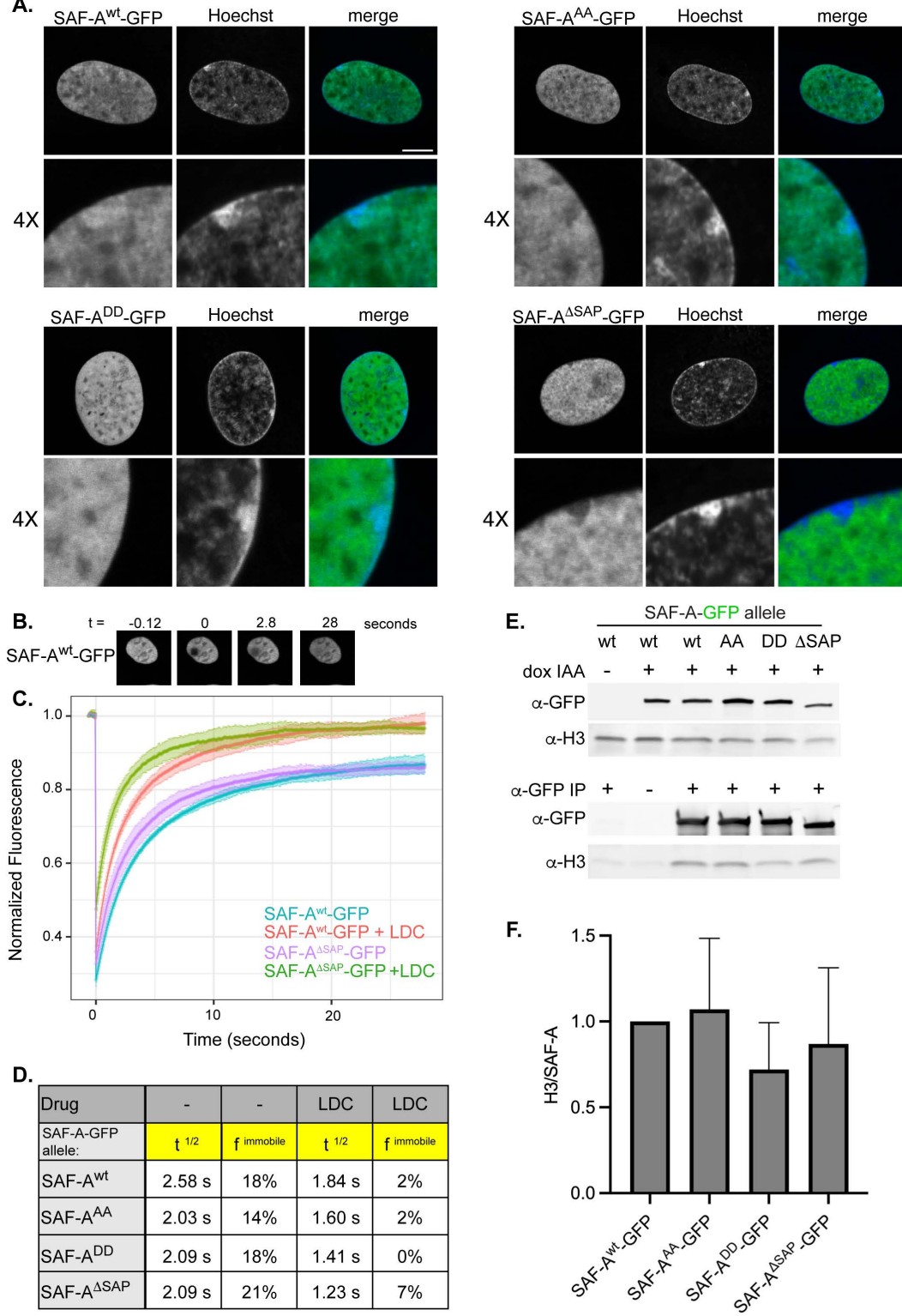

**Fig 3. The SAF-A SAP domain is important for nuclear dynamics and Xi localization.** A. Live cell analysis of SAF-A^wt-GFP and SAP domain mutations as indicated. Cells were analyzed 24 hours after doxycycline and IAA treatment. Each image represents a single 0.2 mm slice. Bar, 10 mm. The proportion of cells showing the depicted localization patterns for each allele are as follows: SAF-A^wt-GFP, 91% of cells; SAF-A^AA-GFP, 96% of cells;

D.

| Drug | - | - | LDC | LDC |
|---|---|---|---|---|
| SAF-A-GFP allele: | $t^{1/2}$ | $f^{immobile}$ | $t^{1/2}$ | $f^{immobile}$ |
| SAF-A^wt | 2.58 s | 18% | 1.84 s | 2% |
| SAF-A^AA | 2.03 s | 14% | 1.60 s | 2% |
| SAF-A^DD | 2.09 s | 18% | 1.41 s | 0% |
| SAF-A^ΔSAP | 2.09 s | 21% | 1.23 s | 7% |

SAF-A$^{DD}$-GFP 100% of cells; SAF-A$^{ΔSAP}$-GFP 100% of cells (n ≥ 50). B. Images from a typical FRAP experiment using SAF-A$^{wt}$-GFP. C. FRAP recovery curves for SAF-A$^{wt}$-GFP, SAF-A$^{AA}$-GFP, SAF-A$^{DD}$GFP, and SAF-A$^{ΔSAP}$-GFP, with and without transcriptional inhibition (LDC). The standard deviation of recovery time is indicated in light colored error bars. D. Table of the t$^{1/2}$ recovery time and the immobile fraction for all SAF-A-GFP proteins, with and without transcriptional inhibition. E–F. Coimmunoprecipitation analysis of SAF-A-GFP alleles and bulk chromatin. E. Western blot analysis of SAF-A-GFP and histone H3 in soluble extracts (top panels) and α-GFP immunoprecipitations (bottom panels). F. Western Blot quantitation of histone H3 in eluates, expressed as a ratio to the amount of SAF-A-GFP in immunoprecipitations. SAF-A$^{wt}$-GFP was normalized to 1.0. The graph depicts the mean and SEM of n = 5 experiments. Statistical pairwise comparisons for each allele were performed using a Wilcoxon signed rank test to determine the p-value, which was ns.

coimmunoprecipitation data suggest that SAP domain mutations may influence protein-RNA or protein-protein interactions more so than interactions of SAF-A with chromatin during interphase.

## Maintenance of Xi gene expression and Xi chromatin structure in SAF-A depleted cells

RPE-1 cells are a differentiated somatic cell type and therefore have an Xi chromosome in the epigenetically stable maintenance phase of X inactivation. During this phase, transcriptional silencing of the Xi can occur even if XIST RNA function is absent, and instead relies on other mechanisms such as DNA methylation to repress transcription [45–47]. We therefore tested whether the role for SAF-A on the Xi was solely epistatic to XIST RNA by monitoring for changes in Xi transcription and chromatin structure.

Recently, the haplotype-resolved genome sequence of human RPE-1 cells was published [48], suggesting that we could examine X-linked gene expression and chromatin structure using high-throughput sequencing techniques. To measure allele-specific expression we used the Personal Allele Caller software which was shown to have the highest accuracy and least mapping bias [49]. Examining RNA-seq data from wild-type RPE-1 cells revealed that the vast major-ity of autosomal genes have equal expression from the maternal and paternal alleles (Fig 4A). In contrast, genes located on the X-chromosome exhibit a strong bias with most expression arising from the 'a' allele. We note RPE-1 cells contain a 73 MB duplication of chromosome 10 translocated to the X chromosome [48,50]. We detected a modest bias in allele-specific expression on chromosome 10 in our RNA sequencing data, suggesting that this translocation occurs onto the Xi. In contrast to most X-linked genes, 2 of 4 genes reported to escape X inactivation exhibited equal expression between both alleles (Fig 4A red points). Although there are likely more genes that escape Xi silencing, we could detect only 4 with SNPs in RPE-1 cells.

To further validate our ability to detect allele-specific differential expression of genes on the X chromosome, we per-formed ATAC-seq [51] and Cut-and-Run [52] for the histone modifications H3K4me3 and H3K27Ac. Consistent with our allele-specific RNA-seq analysis, we found that chromatin accessibility and active chromatin modifications are equally rep-resented on both alleles on all autosomes except chromosome 10, again likely due to the duplication/translocation to the Xi (Fig 4C). In contrast, for the X chromosome we observed a decrease in chromatin accessibility on the 'b' allele (Fig 4C) and a concomitant decrease in active histone marks on the 'b' allele (Figs 4E and S4A). Taken together, the strong allelic bias in gene expression, chromatin accessibility, and active histone marks solely on the X chromosome strongly suggests that allele-specific sequencing is a viable strategy to monitor gene expression from the Xi in human RPE-1 cells.

To determine if SAF-A depletion resulted in reactivation of genes on the Xi, we used edgeR to compare gene expres-sion from the 'a' (active) and 'b' (silenced) alleles in RPE-1 cells with and without SAF-A depletion. We found that genes located on the X chromosome exhibited nearly identical gene expression after 24 hours of SAF-A depletion (Fig 4B). To determine if SAF-A depletion over a longer time frame would reactivate gene expression from the Xi, we performed RNA-seq at 48 and 72 hours after SAF-A depletion. SAF-A depletion did not globally reactivate Xi gene expression under any of these conditions (S4C and S4D Fig). Similarly, none of the SAF-A SAP domain mutants reactivated gene expression on the Xi (S4E-S4H Fig). We also tested whether SAF-A depletion resulted in altered chromatin accessibility or active chro-matin modifications on the Xi, using ATAC-seq and Cut-and-Run for the H3K4Me3 and H3K27Ac histone modifications.

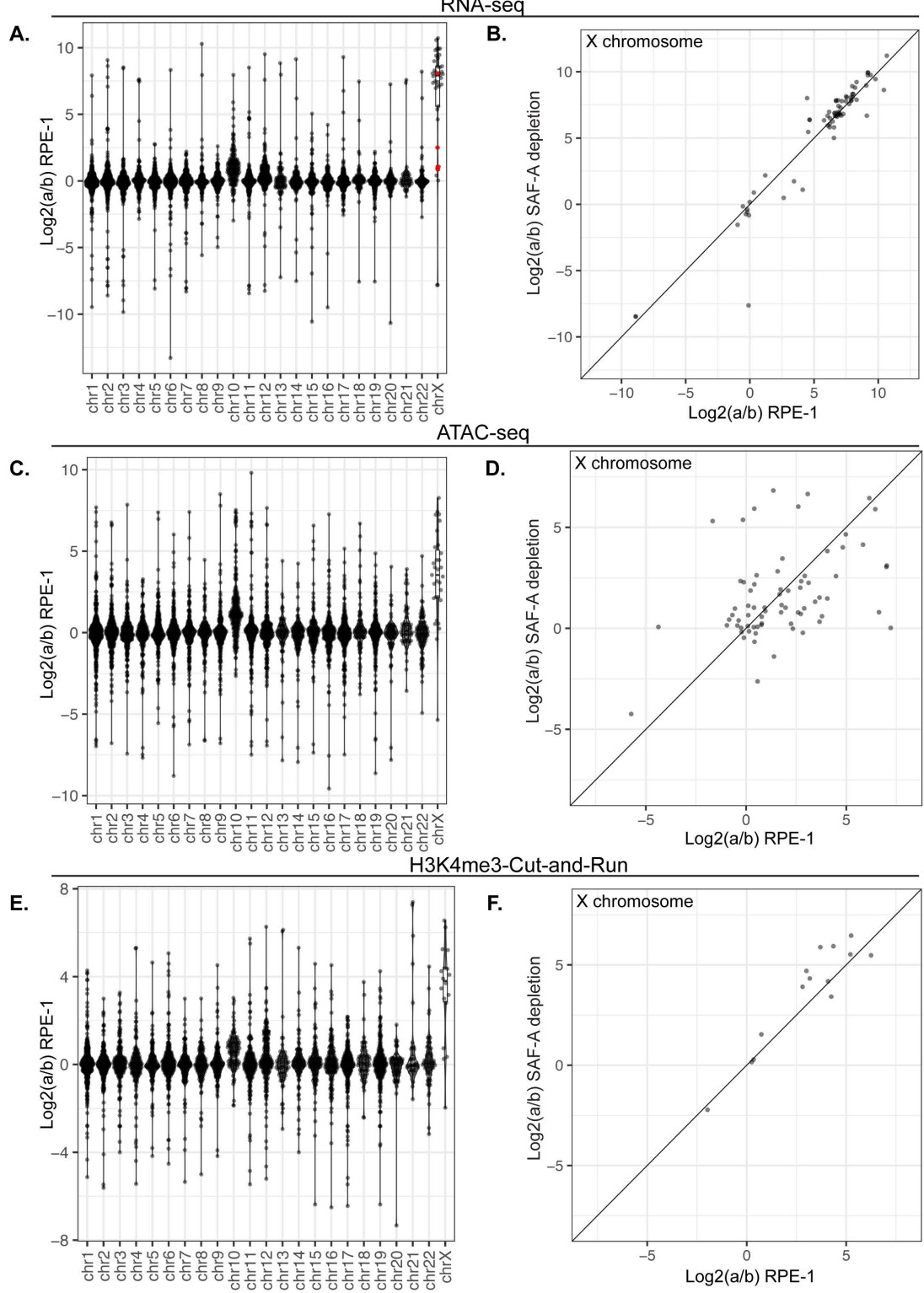

**Fig 4. SAF-A depletion does not reactivate gene expression on the inactive X chromosome.** A. Allele-specific gene expression was calculated from RNA-seq libraries using a combination of PAC and edgeR in RPE-1 cells. 'a to 'b ratios are plotted by gene for each chromosome. B. Average a:b ratio for all genes on the X chromosome plotted for RPE-1 and SAF-A depleted cells. C. Allele-specific ATAC-seq was calculated using PAC and is

plotted by gene for each chromosome. D. comparison of allele-specific ATAC-seq reads by gene plotted for RPE-1 cells and SAF-A depleted cells. E. Allele-specific Cut-and-Run for H3K4Me3 was calculated using PAC. 'a/b' ratio is plotted by gene for each chromosome. F. Allele-specific Cut-and-Run is plotted for RPE-1 cells and SAF-A depleted cells for X chromosome genes.

Consistent with our RNA sequencing results, we did not observe changes in the chromatin structure of the Xi following SAF-A depletion (Figs 4D, 4F, and S4B).

Collectively, the data argue that SAF-A impacts XIST RNA localization and XIST-dependent organization of PRC1 and PRC2 activity on the Xi, but does not affect other parallel pathways enforcing transcriptional silencing in the maintenance phase of X inactivation. This is in contrast to the documented role of SAF-A in establishment of the Xi during cell differentiation [23]. We conclude that the role of SAF-A's contribution to maintaining chromatin structure on the Xi is epistatic to XIST RNA.

## SAF-A depletion results in transcriptional changes that accumulate over multiple cell divisions

To understand how the loss of SAF-A affects steady-state RNA levels over time in RPE-1 cells, we performed RNA-seq on cells depleted for SAF-A for 24, 48, and 72 hours. Interestingly, we observed a gradual increase in the number of misregulated genes during the time course of SAF-A depletion, with a modest number of misregulated genes at 24 hours (227), additional genes misregulated at 48 hours (313) and increasingly more genes misregulated at 72 hours (1196; Fig 5A–5C). At each time point, the majority of misexpressed genes in SAF-A depleted cells exhibited reduced expression relative to control datasets Fig 5A–5C). Comparison of up- and down-regulated genes at each time point revealed a strong overlap between misregulated genes at 24, 48, and 72 hours (Fig 5D and 5E). This suggests that a core subset of genes is misregulated after SAF-A depletion and that additional genes become misregulated after long-term SAF-A depletion, perhaps as a result of cellular responses to a decrease in proliferation. Gene ontology analysis of SAF-A downregulated gene categories over the time course identified a strong enrichment of terms associated with the extracellular matrix, cell adhesion, axon extension, and angiogenesis (Fig 5F). We note the modest effect of SAF-A on gene expression after acute depletion is consistent with a prior report [19], suggesting that the majority of genes misregulated after extended SAF-A depletion are the result of indirect effects. These time-resolved datasets reveal that SAF-A depletion over multiple cell divisions results in cumulative changes in global gene expression.

We tested whether the SAP domain is involved in the role of SAF-A maintaining normal RNA levels (S2 Fig). We performed RNA-seq and compared gene expression in cells expressing SAF-A$^{wt}$-GFP, SAF-A$^{AA}$-GFP, SAF-A$^{DD}$-GFP, and SAF-A$^{\Delta SAP}$-GFP, and SAF-A depleted cells. As in SAF-A depleted cells, we observed only modest changes in gene expression in cells expressing SAF-A$^{AA}$-GFP and SAF-A$^{DD}$-GFP (S5 Fig). Interestingly, gene expression was mostly unaffected in cells expressing SAF-A$^{\Delta SAP}$-GFP, suggesting that interruption of the SAP domain phosphorylation cycle impacts steady-state RNA levels more so than a SAP domain deletion.

## The SAF-A SAP domain plays a minor role in mRNA splicing

Foundational work identified that SAF-A regulates mRNA splicing in multiple organisms and tissue types [8,17,53], through a mechanism involving maturation of the U2 snRNP [53]. Presently, it is not known whether splicing defects in SAF-A depleted cells are emergent at early time points after depletion, or whether the SAP domain is required to promote normal mRNA splicing. To understand how SAF-A contributes to mRNA splicing, we used rMATS to analyze differential splicing in SAF-A depleted cells and all SAP domain mutant cell lines at 24 hours from the mRNA-seq data described above. Consistent with previous work, SAF-A depletion led to changes in the inclusion of ~1600 exons (FDR < 0.01; Figs 6A, 6C, S6A, and S6B). SAF-A depletion tended to result in a loss of exon inclusion, but SAF-A was also required to promote exon exclusion for some transcripts. Gene ontology analysis of overrepresented functional groups exhibiting altered splicing

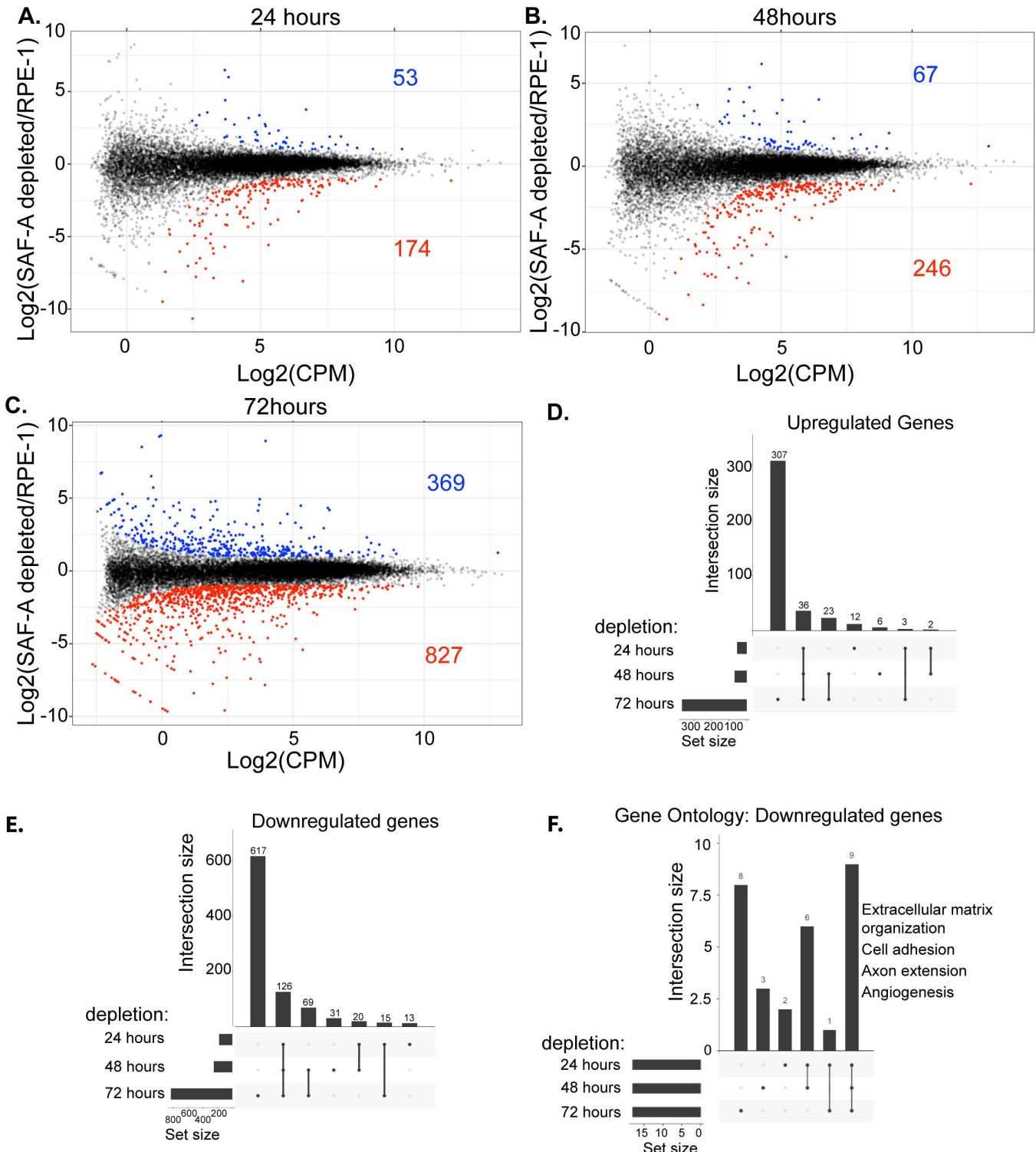

**Fig 5. SAF-A depletion leads to changes in gene expression that accumulate over multiple cell divisions.** A-C. SAF-A was depleted for 24, 48, or 72 hours through addition of auxin. Gene expression was evaluated by RNA-seq and EdgeR. MD plots depict differentially expressed genes at each time point. Genes with an FDR<0.01 are colored. D-E Upset plots depicting the intersection of up and downregulated genes observed at 24, 48, and 72 hours after SAF-Adepletion. F. Gene Ontology analysis of the genes downregulated following acute SAF-A depletion.

revealed that SAF-A is required for the splicing of mRNAs encoding proteins important for transiting the nuclear pore, cell cycle control, and microtubule organization (Fig 6D). Comparison of changes in mRNA splicing to changes in gene expression revealed no correlation, suggesting that changes in mRNAs splicing do not directly cause changes in gene expression (S6C Fig).

To test whether the SAF-A SAP domain was involved in mRNA splicing, we compared splicing changes in cells expressing SAF-A^wt-GFP, SAF-A^AA-GFP, SAF-A^DD-GFP, and SAF-A^ΔSAP-GFP, and SAF-A depleted cells Fig 6B). Comparison of these genotypes revealed that SAF-A depleted cells showed the most significant shift from the wild-type condition (p-value 2.2 x 10^-16 SAF-A^wt-GFP vs. SAF-A depletion), with more minor changes observed in cells expressing SAF-A^ΔSAP-GFP (p-value 1.375 x 10^-8 SAF-A^wt-GFP vs. SAF-A^ΔSAP-GFP, Fig 6C). In contrast, mutation of the SAP domain serines in cells expressing SAF-A^AA-GFP and SAF-A^DD-GFP had very little effect on splicing.

To confirm the observed mRNA splicing changes, we used RT-PCR with primer pairs spanning regulated exons. We tested 4 mRNAs exhibiting both increased and decreased exon inclusion in the SAF-A mutants and confirmed the magnitude and direction of all changes predicted by mRNA sequencing (S7A and S7B Fig). We conclude that SAF-A is important for mRNA splicing and that that the SAP domain plays a minor role in this process.

To understand whether splicing changes accumulate over time after SAF-A depletion, we analyzed splicing changes at 24, 48 and 72 hours. Surprisingly, we did not observe a phenotypic gradient over time with mRNA splicing as we had observed with gene expression. Splicing changes were immediately apparent 24 hours after depletion, and the severity of splicing defects did not increase over time (S8A and S8B Fig). We conclude that acute depletion of SAF-A leads to dramatic changes in splicing, suggesting that SAF-A plays a direct role in this process.

Splicing can be regulated by protein binding to pre-mRNA transcripts at splicing enhancer and repressor sites both within exons, and upstream and downstream of regulated exons [54]. To determine if specific sequence motifs are present in or around SAF-A regulated exons we used MEME [55] and a custom k-mer enrichment program [56] to analyze various regions in and around SAF-A regulated exons. We did not detect significant enrichment of any sequence motifs in any class of sequence (S9A–S9D Fig). To determine if SAF-A binding is enriched in or around SAF-A regulated exons we utilized SAF-A eCLIP data from the ENCODE consortium from two different cell lines. We did not detect any SAF-A enrichment upstream, downstream or in regulated exons in the eCLIP data, consistent with a wide distribution of SAF-A binding throughout the transcriptome (S10A Fig) [57]. In addition to sequence motifs, it is possible SAF-A could recognize secondary structure in mRNA. We used mfold [58] to predict the free energy of control and SAF-A regulated exon regions, but detected no differences related to SAF-A regulation. We also found that SAF-A splicing regulation was not correlated with gene or intron length (S10B–S10D Fig). In contrast, when we compared SAF-A regulated transcripts to RNAs identified as being stably associated with SAF-A [27], we found that transcripts containing exons that exhibited increased inclusion after SAF-A depletion were significantly enriched in SAF-A IP samples (p-value 4.9 x 10^-9, Fig 6E). These data suggest that stable association of SAF-A with a transcript may promote exon exclusion, but that the mechanism targeting SAF-A to specific transcripts is currently unclear. Given the lack of sequence specificity among SAF-A-dependent spliced transcripts, it is possible that splicing defects in SAF-A depleted cells are more attributable to a general defect in the splicing machinery, such as defective U2 snRNP maturation [53].

## Discussion

In this study, we describe how a degron system in diploid, karyotypically stable RPE-1 cells can be used to explore SAF-A function in several genetic assays. Given the high abundance of SAF-A in the nucleus [59], one advantage of this system is that it allows for the rapid depletion of SAF-A, allowing for observation within 24 hours, as well as longer time periods. This allowed us to obtain time resolved datasets for the role of SAF-A in cell proliferation, gene expression, and splicing. The stable integration of inducible SAF-A transgenes allowed for expression to approximate normal protein levels as closely as possible. Matching endogenous SAF-A expression levels is important because overexpression of SAF-A has

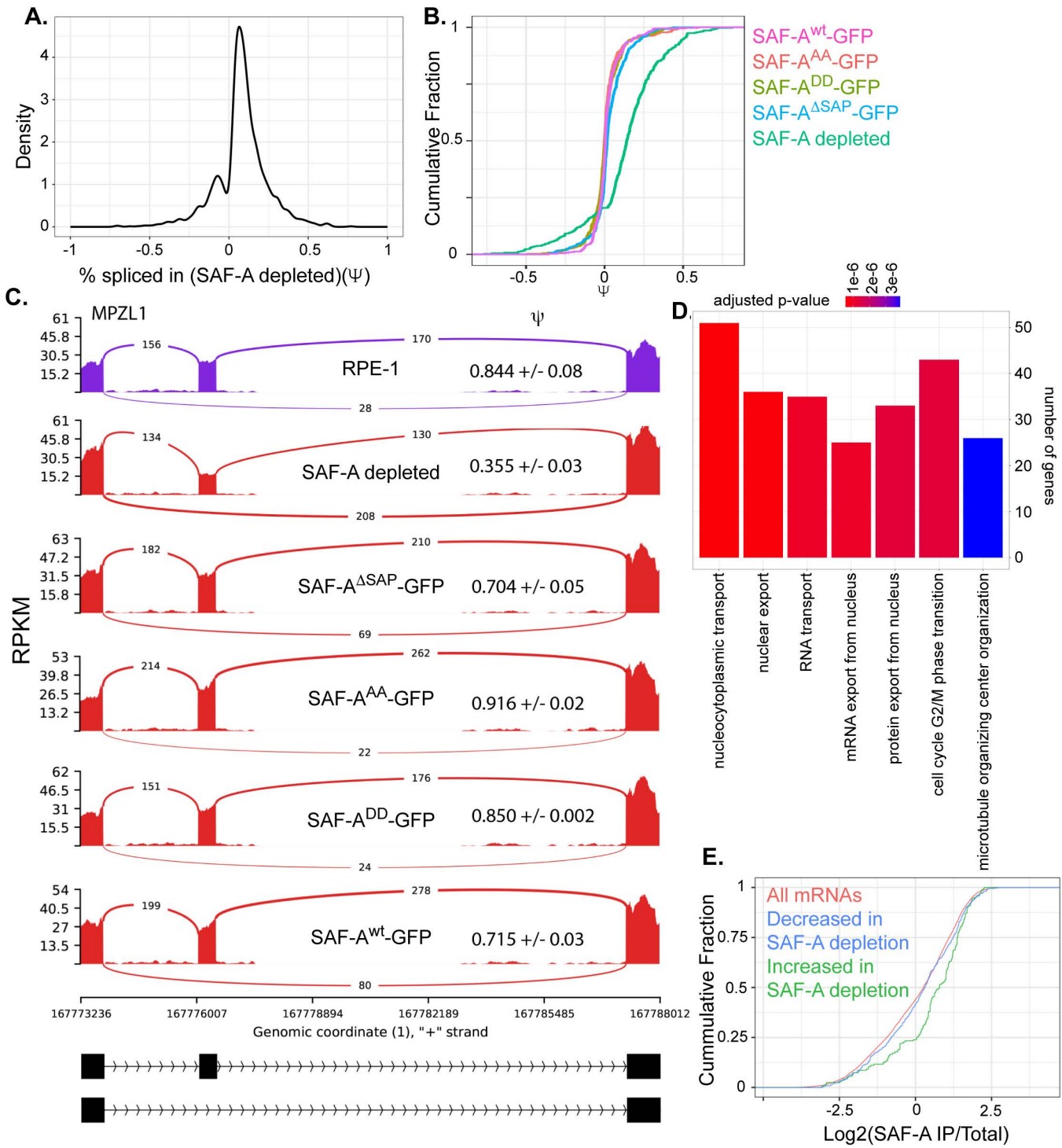

**Fig 6. SAF-A depletion leads to widespread changes in mRNA splicing.** A. mRNA splicing was evaluated in SAF-A depleted cells (24 hours) using rMATS. Density plot showing changes in exon inclusion. B. CDF plot illustrating the magnitude of changes in exon inclusion (of all exons significantly changed following SAF-A depletion) in all domain mutants. C. MISO plot showing an example altered exon. D. GO analysis of genes with altered splicing. E. CDF plot showing enrichment of all mRNAs, mRNAs with increased exon inclusion, and mRNAs with decreased exon inclusion in SAF-A RIP-seq libraries from interphase cells.

been shown repress transcription [60]. In addition, because the SNPs present in RPE-1 cells have been mapped to each haplotype genome, this allowed us to examine X-linked gene expression and chromatin structure with allele-specific resolution. This RPE-1 degron system engineered to simultaneously deplete endogenous SAF-A and induce tagged SAF-A transgenes therefore represents a set of valuable reagents to study SAF-A cellular function.

We used this system to explore the role of the SAF-A SAP domain and SAP domain phosphorylation in supporting XIST RNA localization, applying quantitative image analysis to evaluate XIST RNA localization among several genotypes. Using this assay, we found that a SAP domain deletion is defective for XIST RNA localization to the Xi in human cells, consistent with a prior study in mouse cells [21]. Interestingly, the SAF-A alleles modeling the unphosphorylated (SAF-A^AA-GFP) and phosphorylated state (SAF-A^DD-GFP) had opposite XIST RNA localization phenotypes: SAF-A^AA-GFP had normal XIST RNA localization, and SAF-A^DD-GFP showed a disruption in XIST RNA localization. These phenotypes could be attributable to differences in the affinity of the phosphorylated SAP domain for RNA, or other nuclear structures.Interestingly, both SAF-A^AA-GFP and SAF-A^DD-GFP were defective in maintaining PRC1- and PRC2-dependent histone modifications on the Xi. XIST RNA is required to organize PRC1 and PRC2 activity on the Xi, but multiple studies suggest this occurs in a manner requiring one or more adapter proteins to bridge the interaction, such as hnRNP K or others [39]. Taken together, these data suggest that the dynamic phosphorylation of SAF-A on serines S14 and S26 could regulate formation of a functional, mature XIST RNP. Since SAF-A localizes along the length of XIST RNA [57], and SAF-A interacts with nascent transcripts (Fig 3 and [19], it is possible that SAF-A is interacting with newly transcribed XIST RNA molecules, facilitating contacts with effector proteins along the length of the RNA.

We examined SAF-A nuclear dynamics and found that most of the SAF-A population is engaged with high turnover interactions with nascent transcripts or other nuclear structures. The observation that SAF-A is a highly dynamic protein prompts a re-evaluation of two concepts in the field. First, the data argue that most SAF-A molecules are not part of a static nuclear structure proposed by the scaffold/matrix model of the nucleus [61]. Second, the data suggest that the SAF-A RNP makes rapid, transient engagements with chromatin or other nuclear elements, rather than being a stable tether to bridge RNA molecules to chromatin. Therefore, it is likely that SAF-A molecules bound to relatively immobile lncRNAs such as XIST RNA occur in the context of high turnover interactions [35,62], driven in part by dynamic phosphorylation on SAP domain serines S14 and 26.

We observed that ~20% of the SAF-A population shows relatively immobile nuclear dynamics and is engaged with nascent RNA in a manner independent of the SAP domain (Fig 3D). We note SAF-A was originally isolated as one of the hnRNP proteins bound to nuclear RNA and has a second nucleic acid binding domain on the C terminus that has strong affinity for RNA (Fig 1A, RGG domain; [16,27]). These observations suggest that the RGG domain may have a role in modulating SAF-A nuclear mobility, by determining SAF-A interactions with nascent RNA species.

There are conflicting observations in the literature about whether SAF-A is enriched or depleted on the Xi in fixed cells [42,63]. We observed the modest enrichment of SAF-A^wt-GFP on the Xi only in live imaging experiments, suggesting that there are changes occurring during cell permeabilization or fixation that are not accurately reporting the localization of SAF-A to the Xi. Therefore, we urge caution when interpreting SAF-A localization in fixed cells. Indeed, previous studies investigating localization of transcription factors demonstrated that live imaging was an indispensable tool to distinguish bona fide chromatin interactions from fixation artifacts [64–67]. For this reason, we used live imaging to confirm the enrichment of SAF-A in the Xi territory during interphase (this study), and to confirm that SAF-A is excluded from chromosomes during mitosis [27,68].

The Xi chromosome territory is a distinct membraneless compartment in the nucleus. Although RNA Pol II can readily access Xi chromatin, its engagement with chromatin is reduced relative to the rest of the nucleus [69,70]. In addition, the Xi territory is characterized by concentrated assemblies of proteins nucleated by XIST RNA, including SPEN, PTPB1, MATR3, TBP43, CELF1, CIZ1 and others [35,71–73], resulting in a network of protein assemblies that coordinate chromosome-wide transcriptional silencing. We were surprised to observe that cells expressing the SAF-A^ΔSAP-GFP allele showed marked exclusion from the Xi territory. This stands in contrast to the SAF-A^wt-GFP allele, which was enriched

on the Xi. These data suggest there is a Xi-localization signal within the SAP domain that is largely independent of SAP domain phosphorylation. Further, the imaging data suggests the Xi-targeting sequence present in the SAP domain enables SAF-A to engage with the Xi compartment in a manner distinct from how SAF-A interacts with the rest of the nucleus, possibly due to the reduced transcription on the Xi.

Previous work demonstrated that short-term depletion of SAF-A using siRNAs resulted in very few changes in gene expression [19]. Other studies in mice and human iPS cells found that depletion or mutation of SAF-A results in changes in the expression of hundreds of genes [11–13]. In this study, we used our system to explore the kinetics of gene expression changes in cells lacking SAF-A function and found that gene expression changes accumulate over multiple cell cycles. It is possible that this occurs through an indirect mechanism, such as a misexpressed gene or misspliced RNA causing long term amplification of gene expression changes. It is also possible that SAF-A affects a low turnover element of chromatin, such as a long-lived histone modification or chromatin modifier. Of note, we identified that SAF-A regulates the expression levels of a subset of RNAs that are distinct from those that are subject to SAF-A-dependent splicing, suggesting these processes are controlled by different mechanisms. Consistent with this idea, we found that cells expressing a SAP domain deletion maintained proper gene expression, but displayed splicing defects of some transcripts. Future work is needed to understand how SAF-A maintains the gene expression state, and which other conserved domains contribute to this function.

In contrast to gene expression, the impact of SAF-A depletion on splicing, XIST RNA localization, and XIST RNA-dependent histone modifications was observed within the first cell cycle of depletion, suggesting that SAF-A is directly linked and continually required to execute each of these functions. Importantly, the ability to compare alleles across assays in this study revealed that the phosphomimetic allele, SAF-A$^{DD}$-GFP, represents a separation of function mutation: whereas SAF-A$^{DD}$-GFP has a relatively normal profile for mRNA splicing, SAF-A$^{DD}$-GFP failed to enforce normal XIST RNA localization on the Xi. Therefore, the role of SAF-A in maintaining XIST RNA localization is independent of mRNA splicing. In addition, the SAF-A$^{AA}$-GFP could execute normal XIST RNA localization but was defective in the recruitment of PRC1 and PRC2-dependent histone modifications. We note that localization and silencing activity are separable functions of the XIST RNA and are attributable to different functional repeat sequences on the RNA [74].

Our studies of cell cycle markers show that the SAF-A SAP domain, and SAP domain serines S14 and S26, are required to support normal cell proliferation. These data suggests that dynamic phosphorylation of S14 and S26 is important to fulfill essential roles of SAF-A. It is likely that this role is critical to maintaining rapid cell divisions during early development and could explain why knockout of the mouse Saf-a gene results in early embryonic lethality [8]. In addition, we observed misexpressed genes in a number of categories important for normal organismal function (Fig 5). We conclude the constellation of phenotypes we present here provide insight for SAF-A SAP domain functions at both the cellular and organismal levels.

## Materials and methods

### Cell culture and drug treatment

hTERT-immortalized RPE-1 cells were a gift from Brian Chadwick (Florida State University, Tallahassee, FL; ATCC CRL-4000) and were cultured as described [27]. The SAF-A degron cell line in RPE-1 cells has been described [27]. Lentiviruses with SAF-A$^{wt}$-GFP, SAF-A$^{AA}$-GFP, SAF-A$^{DD}$-GFP, and SAF-A$^{\Delta SAP}$GFP were prepared and used to transduce the SAF-A-degron cell line. Individual clones were expanded and validated by immunofluorescent detection of GFP and western blot analysis.

To induce SAF-A depletion +/- SAF-A-GFP expression, cells were treated with 0.5 µg/mL doxycycline (D3447; MilliporeSigma) and 100 µg/mL 3-indole acetic acid (IAA, I-5148 MilliporeSigma) for 24 hours. Control experiments determined that treatment with 0.5 µg/mL doxycycline resulted in SAF-A-GFP expression levels equivalent to the native, untagged protein. For the long-term depletion of SAF-A, media was replaced with fresh drug every 2–3 days.

## Plasmids

GFP-tagged SAF-A alleles were cloned into the lentiviral expression vector pLVX-TetOne-puro (Takara Bio). The lentiviral plasmids pMB1103 (SAF-A$^{wt}$-GFP), pMB1109 (SAF-A$^{AA}$-GFP), and pMB1311 (SAF-A A$^{DD}$-GFP) have been described [27]. pMB1316 (SAF-A$^{\Delta SAP}$GFP) was prepared similarly.

## Antibodies

Primary antibodies used in this study were as follows: mouse anti-GFP (sc-9996, Santa Cruz Biotechnology), rabbit anti-mCherry (600–401-p16, Rockland), mouse anti-tubulin (DM1A, MilliporeSigma), GFP booster nanobody-ATTO488 (GBA488, Bulldog Bio), RFP booster nanobody-ATTO 594 (RBA594, Bulldog Bio), mouse-anti histones H1-H4 (MAB3422, MilliporeSigma), rabbit anti Ki-67 (ab16667, Abcam), rabbit anti-H2AK119ub (D27C4, Cell Signaling Technology), mouse anti-H3K27me3 (39537, Active Motif), mouse anti-histone H3 (1B1B2, Cell Signaling Technology), and human anti-PCNA (AK, a gift from Dr. Yoshinari Takasaki (Juntendo University School of Medicine, Japan).

Secondary antibodies used in western blotting were as follows: donkey anti-mouse conjugated to 680RD (925–68072, LiCor) and donkey anti-rabbit conjugated to 680RD (925–68073, LiCor). Secondary antibodies used in immunofluorescent studies were donkey anti-mouse Alexa Fluor 647 (715-605-150, Jackson Immunoresearch), donkey anti-human Alexa Fluor 488 (709-545-149, Jackson Immunoresearch), donkey anti-human Cy3 (709-165-149, Jackson Immunoresearch), donkey anti-rabbit Alexa Fluor 488 (711-545-152, Jackson Immunoresearch), donkey anti-rabbit Cy3 (711-165-152, Jackson Immunoresearch), donkey anti-rabbit Alexa Fluor 647 (Jackson Immunoresearch), and donkey anti-mouse Cy3 (715-165-150, Jackson Immunoresearch).

## Western blot analysis and immunoprecipitation

Cells were grown on 15 cm dishes and washed once with 1X PBS. Cells were incubated for 30 min at 4°C in ice-cold lysis/IP buffer (1 ml per 15-cm plate) containing 25 mM Tris, pH 7.4, 150 mM KCl, 5 mM EDTA, 5 mM MgCl$_2$, 1% NP-40, 0.5 mM DTT, and protease inhibitors (Pierce A32955). Cells were collected from the plate and passed several times through a syringe with a 25-gauge needle. Lysates were centrifuged for 30 min at 4°C (~22,000 × $g$) to remove insoluble material. The protein concentration of the extract was determined using a Bradford assay. For each cell extract, 10 µg was loaded onto a 4–20% polyacrylamide gel (Bio-Rad 4561096) and run at 15 mA per gel to resolve the different SAF-A protein isoforms. Proteins were transferred to PVDF membrane and probed with antibodies specified above. Blots were exposed using a LI-COR Odyssey 9120 fluorescent imaging system. Blot quantitation was performed using Bio-Rad ImageQuant software.

To immunoprecipitate SAF-A-GFP, cell extracts were prepared as described above, and protein concentration was adjusted to 1 mg/mL. Extracts were incubated with 25 µL pre-equilibrated bead slurry of GFP-Trap Magnetic Agarose (chromotek GTMA020, Bulldog Bio) or Binding Control Magnetic Agarose (chromotek BMAB020) according to the manufacturer's instructions. After 1 hour incubation at 4°C, immune complexes were collected with a magnet and washed three times with lysis/IP buffer. After the last wash samples were eluted in 15 µL sample buffer and analyzed as described above.

## Image acquisition and analysis

All images were acquired using a Nikon A1R confocal microscope equipped with a 60 × 1.4NA Vc lens and laser lines at 405, 488, 562, and 647 nm. Images were collected using Nikon Elements software driving the galvano scanner. Images were acquired as Z stacks spaced 0.2 µm apart. Each experiment was repeated in at least two biological replicates. For immunofluorescent detection of SAF-A alleles and Ki-67 (Fig 1E and 1G), we used the same fixation and detection conditions we previously described [27]. For PCNA and immunofluorescent detection of histone H3K27me3 and histone

H2AK119ub, we performed permeabilization with CSK buffer (100 mM NaCl, 300 mM sucrose, 3 mM MgCl$_2$, 10 mM PIPES pH 6.8) for 30 seconds, followed by treatment with CSK buffer + 0.5% Triton-X-100, followed by CSK buffer for 30 seconds, and fixation with 4% paraformaldehyde (15710, Electron Microscopy Sciences) buffered with 1X PBS. Our FISH conditions to detect XIST RNA have been published previously [27,75,76]. Images were rendered in Fiji software as a projection or single optical slice, as specified in the figure legends.

### Live imaging and FRAP

For live imaging of SAF-A-GFP alleles on the Xi, cells were plated onto fluorodishes (FD35–100, World Precision Instruments) and treated with doxycyline and IAA for 24 hours prior to observation. Hoechst 33342 dye (Molecular Probes) was added just prior to imaging. Cells were imaged on a Nikon A1R confocal microscope equipped with a stage- top incubator and CO$_2$ chamber (Tokai Hit). For FRAP analysis, cells were plated as above, except without Hoechst dye. FRAP was performed as follows: the scan area was set to 128 x 128, at a scan zoom size of 8X. 10 images were acquired prior to bleaching. Bleach area was a 3 x 3 square. Bleaching was accomplished with 80% power of the 488 laser for 1 second. Images were acquired every 65 ms for 30 seconds following bleaching (456 images). For each allele, FRAP movies were collected from at least 25 cells in total on two different days. For experiments involving the transcriptional inhibitor LDC, the drug was added to the culture media 30 minutes prior to imaging. Movies were imported into Fiji and quantitative analysis of FRAP data was performed as described [77]. FRAP curves were plotted in R from average values across all cells analyzed.

### ATAC-seq

1 million RPE-1 or SAF-A depleted cells were used as input for the Zymo-seq ATAC-seq kit (D5458). Libraries were constructed according to the manufacturer's instructions. Libraries were sequenced to a depth of ~120 million reads at Novogene using paired-end 150 bp reads.

### Cut-and-run

1 million RPE-1 or SAF-A depleted cells were used as input for the EpiCypher CUTANA ChIC/CUT&RUN Kit Version 3 (14–1048), with anti-H3K4me3 (EpiCypher 13–0060), anti-H3K27Ac (Invitrogen, MA5–23516) or IgG antibodies. 5 ng of CUT&RUN enriched DNA was used as input for library construction with the EpiCypher CUT&RUN Library Prep Kit (14–1001). Libraries were sequenced at a depth of ~20 million reads at Novogene using paired-end 150 bp reads.

### RNA-seq

rRNA depleted samples were converted into libraries using the NEB Next Ultra II RNA library kit. Libraries were sequenced by Novogene using 150 bp paired-end reads.

### RNA-seq analysis

**Common QC and processing steps.** All libraries were sequenced at Novogene using paired-end 150 bp reads. Raw fastq files were analyzed for quality, adaptor sequences were trimmed, and duplicate reads were removed using the fastp software suite [78]. Unique, trimmed fastq files were used for all subsequent steps.

**Differential gene expression analysis.** De-duplicated, trimmed reads were aligned to the UCSC hg38 human genome sequence using Rsubread/subread [79]. Reads per gene were quantitated using FeatureCounts in R with Entrez Gene IDs as the reference. Differential gene expression analysis between control and mutant cell lines was performed using edgeR [80]. For the plots in this study we considered genes significant if they exhibited a logFC of>= 1 and FDR value less than 0.01. All MD plots were created using ggplot2 in R.

**Gene ontology analysis.** Significantly altered genes were selected from edgeR analysis. Differential genes were split into up- and down-regulated genes and submitted to EnrichR [81] Gene Ontology analysis using Biological Process and Benjamani Hofberg p-value correction. Enriched categories were manually curated to combine redundant terms and plotted using the enrichR results and ClusterProfiler [82].

**Alternative splicing analysis.** De-duplicated, trimmed reads were used as input for rMATS [83] in conjunction with the ENSEMBL hg38 and ENSEMBL transcript annotations. RMATS output files were filtered as follows: first, we only considered splicing events supported by an average of 12.5 junction spanning reads per sample (e.g., for a comparison including two control and two experimental samples we required a total of 50 reads). For determination of the number of significant changes we filtered for a FDR < 0.01. To produce Sashimi plots we used rmats2sashimi.py. To conduct GeneOntology analysis of alternative splicing changes we used the ENSEMBL Gene names of significantly changed exons as input for EnrichR and ClusterProfiler. For analysis of conserved domains altered by alternative splicing we used SpliceTools and batch blast against the NCBI conserved domain database.

**Motif analysis of alternatively spliced exons.** To search for motifs near alternatively spliced exons we used bedtools to collect 100 bp upstream and downstream of regulated exons and unregulated exons (as a control). We also used bedtools to collect the sequences of all regulated exons and unregulated exons. We then used these sequences as input for STREME for discriminative identification mode. As an alternative approach we used these sequences as input for a custom Perl script that calculates the frequency of all k-mers in a set of experimental and control sequences then derived an enrichment score for each k-mer. For this analysis we analyzed hexamers but obtained similar results for k-mers ranging from 5-8. For secondary structure DeltaG prediction we used all the sequences described above as input for the Vienna RNAfold package. DeltaG values were extracted from the batch output using R and compared between regulated exons and nonregulated exons.

**Allele-specific gene expression analysis.** We used the Personalized Allele Caller [49] software in conjunction with the RPE-1.vcf file [48] of maternal and paternal SNPs to generate counts per gene from each RNA-seq library. To validate the RPE-1.vcf file with our own batch of RPE-1 cells we sequenced the genome of these cells to a depth of 20X using paired-end 150 bp reads. We then ran PAC on this DNA sequencing data using the published.vcf file. We eliminated all SNPs with 0 counts for either allele in the DNA sequencing experiment and all SNPs that exhibited an allele-specific bias of > 1 Standard Deviation from the mean. We used the filtered.vcf file for all subsequent experiments. We filtered the genes to only consider genes with a total of greater than 10 counts per gene. To compare maternal:paternal ratios in different conditions we created average maternal:paternal ratios for each gene from pairs of control and SAF-A depleted/mutant cells. For comparison of RPE-1 to SAF-A depleted cells, this consisted of 6 libraries for each condition. For comparison of RPE-1 to each SAF-A mutant it consisted of 2 libraries for each condition. For the time series of SAF-A depletion, it consisted of 2 libraries for each condition. Averaged ratios were used to create violin plots in R.

**DGE analysis of gene reactivation on the inactive X chromosome.** To determine if genes are reactivated on the inactive X chromosome, we used edgeR to calculate a logFC and FDR value for a:b ratios for each gene on the X chromosome. We then compared the logFC (a/b) and FDR values for different genotypes. The number of libraries compared for each condition is described above.

**ATAC-seq library QC.** Sequencing.fastq files were initially QCed using fastp as described above. Reads were aligned to hg38 using bwa-mem. Output.bam files were analyzed for fragment size and genomic location using ATACseqQC [84]. To perform allele-specific ATAC-seq QCed trimmer, deduplicated fastq files were used as input for PAC with the modified. vcf file described above.

**Cut&Run analysis.** Sequencing fastq files were QCed using fastp as described above and aligned using PAC as described above. Allele specific read counts for IgG were subtracted from the read counts at the same gene in the antibody libraries to remove background. Genes were filtered for>= 0 reads on for each allele and >5 reads total. The ratio of "a" to "b" allele were calculated for each gene and averaged across replicates.

  

## Quantitation of number of XIST particles in SAF-A depleted and SAP mutant cells

3D images of XIST localization in RPE-1, SAF-A depleted cells, and all SAF-A-GFP cell lines were acquired in 0.2µm z stacks. To quantitate the number of XIST RNA particles in z-stack confocal images, we used a macro implemented in FIJI software. Briefly, we used the DAPI signal to create a mask for nuclei in each image slice after performing a Gausian blur, Otsu threshold, fill holes and exclude nuclei touching the image edges. Nuclei were then added to the 3D ROI Manager to create linked 3D objects. We then detected XIST RNA particles by segmenting the RNA FISH image using the Otsu method. XIST particles were then added to the 3D ROI manager. We then used the 3D ROI Manager to measure properties of all particles, measure colocalization between all particles, and measure distances between all particles. Three measurement files were saved for each image. We then used a custom Perl script to summarize the measurements for each nucleus from these three files. Measurements were performed in a batch manner with no user supervision. The FIJI macro and Perl scripts are listed in S1 and S2 Files.

## Supporting information

**S1 Fig. Expression of SAF-A tagged and SAP domain mutant isoforms.** A. Cells were scored for SAF-A-AID-mCherry expression after 24 hours of drug treatment. The graph depicts the average percent of cells (n = 100) expressing the degron allele; error bars depict the SD for two biological replicates. B. Western blot analysis with the mouse anti-SAF-A 3G6 monoclonal antibody to compare levels of endogenous SAF-A (lane 1), SAF-A-AID-mCherry -/+ drug treatment (lanes 2 and 3) and SAF-A$^{wt}$-GFP (lane 4). C. Quantitation of SAF-A expression levels relative to the tubulin loading control in three different extract preparations. Error bars depict SD. D. MD plot comparing gene expression between wt RPE-1 cells and RPE-1 cells with the SAF-A-AID-mCherry expressed from the endogenous locus. Differentially expressed genes are highlighted in red and blue. E-F. Upset plots comparing the intersection of misregulated genes in untreated SAF-A-AID-mCherry cells to cells depleted of SAF-A by the addition of dox and IAA. G. Quantitation of SAF-A-GFP expression levels. We were unable to identify an antibody that recognizes all tagged versions of SAF-A and SAP domain mutations. Instead, we monitored expression levels of GFP-tagged SAP domain mutations using a GFP antibody and compared to the SAF-A$^{wt}$-GFP control to infer approximate expression levels relative to the endogenous protein. We estimate expression levels of all GFP tagged transgenes is within a 2-fold difference compared to the endogenous protein. (TIF)

**S2 Fig. Characterization of XIST expression and Polycomb-dependent histone modifications in SAF-A depleted cells.** A. MD plot comparing gene expression in 6 replicates of SAF-A depleted cells to wild type RPE-1 cells. XIST RNA expression is highlighted in orange. B. Table depicting the logFC and FDR of XIST RNA for all SAF-A depletion and mutant experiments as measured by RNA-seq. C. Immunofluorescence of H3K27me3 and H2AK119ub in SAF-A$^{wt}$ and SAF-A depleted cells. Scale bar is 10 µm. (TIF)

**S3 Fig. Validation of CDK9 inhibitor LDC.** Cells were incubated with the CDK9 inhibitor LDC, or DMSO, for 30 minutes prior to the addition of EU. Nuclear EU fluorescence intensity was measured from two biological replicates. (TIF)

**S4 Fig. Analysis of X-linked chromatin and gene expression in SAF-A mutants.** A. Allele-specific Cut-and-Run was performed with H3K27Ac antibodies and calculated using PAC. 'a/b' ratios are plotted for each gene by chromosome. B. Comparison of 'a/b' ratios for X linked genes in SAF-A depleted cells. C. Comparison of 'a/b' ratios for all X-linked genes after 48 hours of SAF-A depleted cells. D. Comparison of 'a/b' ratios for all X-linked genes after 72 hours of SAF-A depletion. E-H. Comparison of 'a/b' ratios for all X-linked genes after in all SAF-A SAP domain mutants. (TIF)

**S5 Fig. Gene expression profiles in cells expressing SAF-A SAP domain mutations.** A-E. Gene expression was evaluated at 24 hours after addition of doxycycline and auxin, using RNA-seq and EdgeR. MD plots depict significantly differentially expressed genes for each mutant (FDR<0.01). The gene expression profile of cells depleted for SAF-A, or expressing SAF-A alleles SAF-A^wt^-GFP, SAF-A^AA^-GFP, SAF-A^AA^-GFP, or SAF-A^ΔSAP^-GFP were compared to RPE-1 cells as indicated on the y axis.
(TIF)

**S6 Fig. MISO plots of additional regulated exons in each SAF-A mutant.** A-B. MISO plots of SE events in WT RPE-1 cells, SAF-A depleted cells, and SAP domain mutants. C. Scatterplot comparison of changes in gene expression at 24h to changes in mRNA splicing.
(TIF)

**S7 Fig. Validation of predicted splicing changes using RT-PCR.** A. Illustration depicting included and skipped exons, and PAGE gel analysis of exon inclusion for 4 different genes with predicted changes in exon inclusion in SAF-A depleted cells. B. Quantitation of percentage spliced in in each mutant from three biological replicates.
(TIF)

**S8 Fig. Splicing defects do not increase with time of SAF-A depletion.** A. PCA analysis of skipped exons (SE) in SAF-A depleted cells, RPE-1, and SAF-A depleted cells rescued with SAF-A^wt^-GFP at each time point. B. CDF analysis of SE altered in SAF-A depleted cells at 24 hours in each cell type at each time point.
(TIF)

**S9 Fig. Motif analysis of SAF-A regulated exons.** A. STREME analysis of upstream and downstream intronic sequences surrounding SAF-A regulated exons. B. Hexamer enrichment in the same sequences as in A. C-D. STREME and hexamer analysis of sequences in and surrounding SAF-A regulated exons. Sequence sets tested are indicated on the plots.
(TIF)

**S10 Fig. Analysis of SAF-A eCLIP data surrounding regulated exons.** A. ENCODE project eCLIP data was analyzed using Deeptools in both HepG2 and K562 cells at SAF-A regulated exons (increased or decreased) and at unregulated genes. Average coverage and heatmap are depicted for all datasets. B. mFold calculation of ΔG for indicated sequenes. C-D. Calculated lengths of indicated sequence groups.
(TIF)

**S1 File. XIST3D.ijm.** ImageJ macro used to identify XIST foci in wild-type RPE-1 and SAF-A mutant cells.
(IJM)

**S2 File.** 3DNuclearParticlesNuclearVolume.pl. Perl script used to process the output files from the ImageJ macro in S1 File.
(PL)

**S1 Data. Quantitation of SAF-A expression following IAA addition, quantitation of SAF-A levels after depletion and rescue.**
(XLSX)

**S2 Data. Quantitation of percentage of cells positive for Ki67 and PCNA following SAF-A depletion and rescue.**
(XLSX)

**S3 Data. Quantitation of co-immunoprecipitation of Histone H3 with various SAF-A mutants.**
(XLSX)

**S4 Data. Quantitation of number of XIST spots per cell from all SAF-A genotypes.**
(TXT)

**S5 Data. Quantitation of percentage of cells positive for H3K27Me3 or H2AK119Ub in all SAF-A genotypes.**
(XLSX)

**S6 Data. Normalized FRAP measurements for each SAF-A genotype with and without transcription inhibition.**
(TXT)

**S7 Data. Quantitation of normalized nuclear intensity of EU with and without transcription inhibition with LDC.**
(TXT)

**S8 Data. Quantitation of exon skipping or inclusion by RT-PCR in all SAF-A genotypes.**
(XLSX)

## Acknowledgments

The authors thank Reito Watanabe for helpful suggestions. We thank Chi Jing and Nelson Lau for preparation of the RPE-1 genome sequencing library. We thank Daniel Cifuentes for use of the LiCor fluorescent imager. We acknowledge that much of the computational work reported on in this paper was performed on the Shared Computing Cluster which is administered by Boston University's Research Computing Services.

## Author contributions

**Conceptualization:** Judith A Sharp, Michael D. Blower.

**Data curation:** Michael D. Blower.

**Formal analysis:** Judith A Sharp, Emily Sparago, Michael D. Blower.

**Funding acquisition:** Michael D. Blower.

**Investigation:** Judith A Sharp, Emily Sparago, Rachael Thomas, Kaitlyn Alimenti, Wei Wang, Michael D. Blower.

**Methodology:** Judith A Sharp, Rachael Thomas, Michael D. Blower.

**Project administration:** Michael D. Blower.

**Resources:** Judith A Sharp, Michael D. Blower.

**Software:** Michael D. Blower.

**Supervision:** Michael D. Blower.

**Validation:** Judith A Sharp, Michael D. Blower.

**Visualization:** Judith A Sharp, Rachael Thomas, Kaitlyn Alimenti, Wei Wang, Michael D. Blower.

**Writing – original draft:** Judith A Sharp, Michael D. Blower.

**Writing – review & editing:** Judith A Sharp, Emily Sparago, Rachael Thomas, Kaitlyn Alimenti, Michael D. Blower.

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
