## [Decision Letter · Decision Letter 0]

PGENETICS-D-24-01290

Nuclear compartments have distinct requirements for the SAF-A SAP domain

PLOS Genetics

Dear Dr. Blower,

Thank you for submitting your manuscript to PLOS Genetics. Your manuscript has been evaluated by three reviewers who were all positive, but felt the manuscript required further revision. The reviewers have provided a list of corrections that should be addressed and which should improve the manuscript. One reviewer suggested that the title was too generalised, so please consider modifying the wording.

Please submit your revised manuscript within 60 days Feb 18 2025 11:59PM. If you will need more time than this to complete your revisions, please reply to this message or contact the journal office at plosgenetics@plos.org. Please include the following items when submitting your revised manuscript:

We look forward to receiving your revised manuscript.

Kind regards,

Nick Gilbert

Guest Editor

PLOS Genetics

Wendy Bickmore

Section Editor

PLOS Genetics

Aimée Dudley

Editor-in-Chief

PLOS Genetics

Anne Goriely

Editor-in-Chief

PLOS Genetics

**Journal Requirements:**

At this stage, the following Authors/Authors require contributions: Judith A Sharp, Emily Sparago, Rachael Thomas, Wei Wang, and Michael Blower. Please ensure that the full contributions of each author are acknowledged in the "Add/Edit/Remove Authors" section of our submission form.

The list of CRediT author contributions may be found here: https://journals.plos.org/plosgenetics/s/authorship#loc-author-contributions

3) We noticed that you used the phrase 'unpublished data' in the manuscript. We do not allow these references, as the PLOS data access policy requires that all data be either published with the manuscript or made available in a publicly accessible database. Please amend the supplementary material to include the referenced data or remove the references.

5) We notice that your supplementary Figures, and Tables are included in the manuscript file. Please remove them and upload them with the file type 'Supporting Information'. Please ensure that each Supporting Information file has a legend listed in the manuscript after the references list.

6) We note that your Data Availability Statement is currently as follows: "All genomics data in this study is available at GEO". Please confirm at this time whether or not your submission contains all raw data required to replicate the results of your study. Authors must share the “minimal data set” for their submission. PLOS defines the minimal data set to consist of the data required to replicate all study findings reported in the article, as well as related metadata and methods (https://journals.plos.org/plosone/s/data-availability#loc-minimal-data-set-definition).

- The points extracted from images for analysis..

7) Please ensure that the funders and grant numbers match between the Financial Disclosure field and the Funding Information tab in your submission form. Note that the funders must be provided in the same order in both places as well. State the initials, alongside each funding source, of each author to receive each grant. For example: "This work was supported by the National Institutes of Health (####### to AM; ###### to CJ) and the National Science Foundation (###### to AM)." State what role the funders took in the study. If the funders had no role in your study, please state: "The funders had no role in study design, data collection and analysis, decision to publish, or preparation of the manuscript.".

**Reviewers' comments:**

Reviewer's Responses to Questions

**Comments to the Authors:**

Reviewer #1: Overall this is an interesting manuscript which highlights the role of the SAP domain of HNRNPU in XIST RNA localisation and RNA splicing. The experiments are in general well controlled and executed. There are some areas where better representation of the existing literature would allow the reader to see a better context for the new data.

1. HNRNPU is the official gene name and the name used in databases so it makes the article very inaccessible to persist in using the terminology SAF-A. To broaden access of this article it is important to include HNRNPU in both the title and abstract, otherwise many searches from scientists who came across this protein through a database would miss your article.

2. Fig. 1d . It would be helpful to understand the extent of constitutive degradation caused by addition of the mini-Aid to the HNRNPU ORF by probing the Western blots with anti-HNRNPU antibody as well so we can see to what extent the – auxin condition is reflective of WT cells and to what extent the GFP fusions used for complementation are overexpressing HNRNPU relative to normal endogenous levels. Whilst I recognise it is challenging to mimic endogenous levels, overexpression may itself cause phenotypes (see for example Table 1 in PMID: 10490622). So, it would be helpful for the reader to understand any limitations of the system being used. Whilst the authors state they monitored transgene levels after induction with tet, it would be nice to see the actual data. Especially when they make this statement in the discussion “. The stable integration of inducible SAF-A transgenes allowed for precise titration of SAF-A expression level, in order to approximate normal protein levels as closely as possible.” I note also that in the original paper describing the generation of the RPE HNRNPU-AID cell line again no Western was provided showing HNRNPU levels probed with a HNRNPU antibody compared to the native RPE cells, only +/- auxin conditions of the engineered cell line. Finally I note for the RNAseqs the authors used RPE vs HNRNPU depletion as their two conditions, rather than -/+ auxin. Was this because the -auxin condition already had substantial depletion of HNRNPU due to the well-known constitutive degradation caused with some proteins tagged with auxin degrons?

3. Fig. 1h . Are the DD, AA and delta SAP mutants in fact acting in a dominant negative manner since these cells seem to perform worse in both assays than the HNRNPU depleted condition?

4. In discussing the data presented in figures 3b-d it would probably be worth mentioning that HNRNPU was defined as a HNRNP protein, all of which bind nascent, intronic RNA, moreover, a key RNA binding domain of HNRNPU is the C-terminal RGG box, in fact the phrase was first termed on the basis of its discovery in HNRNPU (PMID:1628625). Therefore, the observation that “there is a population of SAF-A molecules that are stably associated with nascent transcripts independently of the SAP domain (Figure 3d)” is consistent with the prior literature, including the prior work of these authors.

5. Fig. 5. RNA-seq does not measure transcription. It measures the steady state levels of the transcriptome which can be influenced by multiple factors including transcription and decay rates. The text should reflect this and not infer effects on transcription.

6. Fig S4 what do I and S mean? Perhaps a cartoon showing where the primers are on the gene structure and what is being amplified would be helpful for the reader.

7. The discussion comments about HNRNPU being transiently associated with nuclear elements including RNA should be discussed in the context of its well known role as a HNRNP protein, which bind heterogeneous nuclear RNA i.e nascent RNA, which is, in general, chromatin associated via the transcribing polymerase it is exuding from. Furthermore, it has previously been shown that HNRNPU depletion leads to increased levels of U2 snRNP in the cell PMID: 22325991 and that this could account for the major changes in splicing seen following HNRNPU depletion. Especially given the binding sites for HNRNPU on pre-mRNA correlate poorly with splicing changes, implying a general effect on the splicing machinery is at play as suggested in PMID: 22325991. Such a conclusion is consistent with the data presented here in Fig. S7 and should be considered in the discussion.

Minor points:

1. Should LDC be described as a CDK9 inhibitor rather than a polII inhibitor

2. p.14 line 23- Refers to Figure 5e, there is no figure 5e.

P12 line 23- "normal" RPE-1 cells feels like a vague description, maybe qualify with "diploid" or "karyotypically normal" something to that effect. They use "normal, diploid" in the discussion

3. p.17 line 26- close bracket "(Figure 3 and [20],"

4. Supplemental figures:

-inconsistent bolding of first "A." left over from bolding title in 4 and 7

Supplemental figure 1 legend- "C.F" should be "C-F" to match formatting of other legends

Supplemental figure 5- More descriptive/legible names for the figure legend would help readability, at least some underscores or dashes

Supplemental figure 6 legend- replace "C. C-D" with "C-D"

5. Methods- Can't find a description of how they've lysed their cells for the knock down validation western blot in results or methods.

6. References

p.28:There seems to be multiple places where fragments of the competing interests are inserted into references, see 3. 11. and 13. p.31

p.8:

line 8- Reference 17 doesn't seem relevant to the statement (unless I'm mistaken).

Reviewer #2: In the manuscript ‘Nuclear compartments have distinct requirements for the SAF-A SAP domain’ the authors aim to characterise the effect of the N-terminal SAP domain on the functionality of SAF-A by generating a homozygous SAF-A-mCherry auxin mediated degron cell line in human RPE cells, and then complementing these cells with stable heterozygous integration of either WT SAF-A-GFP or a SAF-A mutant-GFP to the TRE3Gs locus. Expression of either a SAP domain deletion, a SAP AA mutant or a SAP DD phosphomimetic mutant is induced with the use of doxycycline.

The authors present limited novel findings in this manuscript: They recapitulate the previous finding that depletion of SAF-A has an effect upon splicing, but none of the SAP domain mutants demonstrate this phenotype. Furthermore, transcriptomic analysis shows minimal effects of SAF-A depletion or expression of mutants unless cells are maintained for 48 or 72 hrs in the absence of SAF-A. As the authors demonstrate in Figure 1 that by this time such cells already exhibit significant cellular stress with 20-40% of cells in G0, and a clear decrease in replication, any analysis of transcription is likely to be unreliable. Likewise, any comparison of genes affected by splicing with genes that show aberrant transcription after 48 or 72hrs is unreliable for the same reason.

A previous publication demonstrated that the mouse SAP domain was required for Xist localisation. A further publication identified Ser14 and 26 in the SAP domain of SAF-A as important during mitosis. The novel data in this manuscript relates to the recruitment of SAF-A and Xist to the inactive X in interphase human cells and its reliance upon dynamic phosphorylation of S14 and S26. This information is a useful contribution to the field, but there are major points which need to be addressed:

Further major points:

1) The authors suggest that there is ‘complete knockdown of endogenous SAF-A in the majority of cells after 24hrs’. Western blot analysis showing the full degree of knockdown after 24hrs should be shown, particularly as a proportion of cells can be seen to express SAF-A mCherry even after 24hr IAA treatment in Figure 1e.

This is of further importance in Figure 1h, where complementation with the panel of mutant transgenes has more effect upon %PCNA positive cells and %Ki-67 positive cells than depletion of SAF-A in its entirety. This might be explained if there were a dominant negative effect on residual endogenous SAF-A mCherry.

2) The authors state multiple times that transgene expression is equivalent to endogenous expression levels, but this is not shown (‘unpublished data’ in methods). This is an important omission from both the western blot and IF data.

3) In figure 2 the authors use analysis of 3D confocal stacks of Xist foci. They suggest that as they are unable to resolve Xist RNA particles normally localised to the Xi with confocal microscopy, an increase in particles distal to the Xi would correlate to more particles that they can resolve. This approach is flawed as it does not distinguish between increased Xist molecules per nucleus versus the same number of Xist molecules that are more dispersed. Furthermore, in Figure 2a, delocalised Xist can be seen in examples provided, but if the images are representative they must be highly thresholded, for example in SAF-A depleted cells, the average Xist foci in the graph is 12 (range approx. 4-30), but approx. 35 foci are evident by eye! Likewise, the idea that an average of 4 Xist RNA foci are evident in the SAF-A WT cells is clearly so inaccurate that the axis should be re-labelled Xist aggregations.

4) The authors do not present data addressing whether SAF-A GFP mutants are equally capable of binding chromatin as endogenous SAF-A. Even the FRAP data in Figure 3c/d does not address this as the dynamics of the GFP transgenes is not compared to that of endogenous SAF-A with an mCherry tag. Without further analysis of general chromatin binding, they cannot make definite claims that recruitment to the Xi is specifically affected.

5) The authors suggest there is a ‘Xi-targeting signal or sequence in the SAP domain, that is largely independent of Ser 14 and 26’, and ‘distinct from other nuclear territories’, but provide no evidence for such sequence. Targeting might be mediated by protein structure and / or interaction with an accessory protein and no other functionality that requires chromatin binding is addressed.

6) The authors cannot argue that ‘dynamic phosphorylation is required for the formation of a functional, mature Xist RNP’ as they present no evidence to suggest that Xist RNP formation is aberrant in any fashion, only that there is a defect in its recruitment / retention.

Minor points:

1) Details of the macro used for analysis of Xist particles should be provided to enable readers to reproduce experiments

2) Figure 2c. Please include the SAF-A WT control and SAF-A depleted images that correlate to data the graph in Figure 2d.

3) Figure 2c/d. Please correlate uH2A / K27 enrichment with an Xi marker – otherwise this data is based on educated guesswork as to where the Xi might be.

7) The accompanying text to Figure 3a suggests there is enrichment of the SAF-A DD mutant on the Xi, but to a lesser extent to the SAF-A AA mutant. There is not enrichment evident in the example. Furthermore, it would be useful for the proportion of cells in which SAF-A Xi enrichment / exclusion is observed to be detailed.

8) Figure 5. Please explain further why interruption of SAP phosphorylation would have more of an effect than deletion of the whole SAP domain on gene expression?

9) When RNA-seq data from the RPE cells is analysed based on haplotype analysis in Figure 4 how many genes are able to be analysed on the X compared to the autosomes with this SNP based approach – it would appear far fewer on the X chromosome versus autosomes.

10) The title is too generalised. The authors have not to my mind demonstrated that ‘nuclear compartments have distinct requirements for the SAF-A SAP domain’.

Reviewer #3: This study focuses on the SAF-A protein’s SAP domain and its phosphorylation in nuclear processes. Using a degron-based system in human RPE-1 cells, the authors show that SAF-A is highly dynamic, transiently interacting with chromatin, RNA, and other nuclear structures. They identify phosphorylatable serines (S14, S26) in the SAP domain as critical for maintaining XIST RNA localization on the inactive X chromosome, enabling PRC1/PRC2-dependent histone modifications, and ensuring normal cell proliferation. Although depleting SAF-A affects mRNA splicing, the SAP domain itself appears to have limited influence on this process. Overall, these findings suggest that SAP domain phosphorylation drives rapid SAF-A turnover in distinct nuclear compartments, providing new insights into regulatory layers of nuclear organization and function. The manuscript is generally well written and concise, with data that largely support the authors’ conclusions. However, the lack of page, line, and figure numbers complicates the review process.

The results will interest researchers in heterochromatin and RNA biology, even though the SAP domain’s physiological impact appears modest in RPE-1 cells. With a few specific issues addressed, this manuscript should be suitable for publication in PLOS Genetics.

Major Points:

1. The authors claim complete endogenous SAF-A depletion with auxin and doxycycline but only show Western blots using anti-fluorescent protein antibodies, not antibodies against endogenous SAF-A. Additionally, there is no genotyping data confirming both alleles are tagged with no remaining untagged allele. To meet PLOS Genetics standards, the authors should provide an antibody that recognizes endogenous SAF-A (if available) and present molecular validation of the knock-in strategy.

2. In Figure 2, the conclusion that Xi targeting is defective is based on a more dispersed XIST signal. The authors should rule out the possibility that this dispersion is due to increased XIST levels rather than mislocalization.

3. The authors use LDC to block transcription but do not include controls confirming effective transcriptional inhibition. Such a control should be provided.

4. The authors mention that gene expression changes accumulate over time, but they do not show whether the same sets of genes are affected at different timepoints. Demonstrating overlap in misregulated genes would strengthen their conclusions.

5. The authors frequently cite figure panels out of sequence (e.g., referring to panel c before panel b). Citing panels in order would improve readability and clarity.

Minor Points:

• In Fig. 2d, the authors should provide statistical analyses.

• In the FRAP experiments, why authors do not examine SAF-A dynamics specifically at the Xi? It could provide direct evidence of distinct turnover in this compartment

• In Fig. S2, the authors should include the number of misregulated genes, as done in Fig. 5.

• The terminology for SAF-A degration (e.g., “depletion” vs. “knockdown”) should be used consistently throughout the manuscript to avoid confusion.

**Have all data underlying the figures and results presented in the manuscript been provided?**

Reviewer #1: Yes

Reviewer #2: Yes

Reviewer #3: Yes

PLOS authors have the option to publish the peer review history of their article (what does this mean? ). If published, this will include your full peer review and any attached files.

**Do you want your identity to be public for this peer review?** For information about this choice, including consent withdrawal, please see our Privacy Policy .

Reviewer #1: No

Reviewer #2: No

Reviewer #3: No

**Figure resubmission:**
---

## [Decision Letter · Decision Letter 1]

Dear Dr Blower,

We are pleased to inform you that your manuscript entitled "Role of the SAF-A/HNRNPU SAP domain in X chromosome inactivation, nuclear dynamics, transcription, splicing, and cell proliferation" has been editorially accepted for publication in PLOS Genetics. Congratulations!

Yours sincerely,

Nick Gilbert

Guest Editor

PLOS Genetics

Wendy Bickmore

Section Editor

PLOS Genetics

Aimée Dudley

Editor-in-Chief

PLOS Genetics

Anne Goriely

Editor-in-Chief

PLOS Genetics

Comments from the reviewers (if applicable):

Reviewer's Responses to Questions

**Comments to the Authors:**

Reviewer #1: The manuscript is substantially improved and the authors have addressed all my original concerns.

I noted a single typo in the Abstract:

" We found that that a SAP domain deletion"

Reviewer #2: I am satisfied that the authors have answered my questions adequately and appreciate the effort they have taken in providing additional data where required.

Reviewer #3: I have read the revised version of the manuscript and appreciate the authors’ detailed and constructive responses to the comments raised during the initial review. In particular, I had requested further validation of the SAF-A depletion strategy. Specifically, I asked for evidence of endogenous SAF-A depletion using antibodies that recognize the untagged protein, as well as molecular confirmation of homozygous tagging. The revised manuscript addresses this point thoroughly: the authors now present Western blot analyses using the 3G6 monoclonal antibody, which detects both endogenous and tagged SAF-A isoforms. This analysis demonstrates successful homozygous targeting and efficient degradation of SAF-A-AID-mCherry following auxin treatment. These data, now included in Supplemental Figure 1, substantially strengthen the manuscript by validating the degron system.

I also raised the possibility that the observed XIST mislocalization upon SAF-A depletion or mutation could be due to increased XIST RNA levels rather than mislocalization. The revised manuscript now explicitly presents XIST expression levels from RNA-seq datasets in Supplemental Figure 2. These data show that XIST expression is unchanged, supporting the conclusion that the phenotype reflects mislocalization rather than transcriptional upregulation.

In addition, the authors now provide appropriate controls for transcriptional inhibition by LDC treatment, using EU incorporation assays included in Supplemental Figure 3. This control validates the effectiveness of transcriptional inhibition and supports the conclusions drawn from their FRAP experiments regarding the link between SAF-A dynamics and transcriptional activity.

Beyond these points, the authors have made numerous improvements throughout the manuscript in response to feedback from all reviewers.

Overall, the authors have satisfactorily addressed all major concerns, and the revised manuscript now meets the standards for publication in PLOS Genetics.

**Have all data underlying the figures and results presented in the manuscript been provided?**

Reviewer #1: Yes

Reviewer #2: Yes

Reviewer #3: Yes

PLOS authors have the option to publish the peer review history of their article (what does this mean? ). If published, this will include your full peer review and any attached files.

**Do you want your identity to be public for this peer review?** For information about this choice, including consent withdrawal, please see our Privacy Policy .

Reviewer #1: No

Reviewer #2: No

Reviewer #3: No

**Data Deposition**

http://datadryad.org/submit?journalID=pgenetics&manu=PGENETICS-D-24-01290R1

**Press Queries**

---

## [Editor Report · Acceptance letter]

PGENETICS-D-24-01290R1

Role of the SAF-A/HNRNPU SAP domain in X chromosome inactivation, nuclear dynamics, transcription, splicing, and cell proliferation

Dear Dr Blower,

We are pleased to inform you that your manuscript entitled "Role of the SAF-A/HNRNPU SAP domain in X chromosome inactivation, nuclear dynamics, transcription, splicing, and cell proliferation" has been formally accepted for publication in PLOS Genetics! Your manuscript is now with our production department and you will be notified of the publication date in due course.

With kind regards,

Anita Estes

PLOS Genetics

On behalf of:
